# GROUP REPRESENTATIONAL POSITION ENCODING

**Yifan Zhang**[1]   **Zixiang Chen**[*2]   **Yifeng Liu**[*2]   **Zhen Qin**[*]   **Huizhuo Yuan**[2]
**Kangping Xu**[3]   **Yang Yuan**   **Quanquan Gu**[2†]   **Andrew Chi-Chih Yao**[3†]

[1]Princeton University    [2]University of California, Los Angeles
[3]IIIS, Tsinghua University

yifzhang@princeton.edu   qgu@cs.ucla.edu
andrewcyao@tsinghua.edu.cn

## ABSTRACT

We present **GRAPE** (**G**roup **R**epresent**A**tional **P**osition **E**ncoding), a unified framework for positional encoding based on group actions. GRAPE brings together two families of mechanisms: (i) *multiplicative* rotations (Multiplicative GRAPE) in $\mathrm{SO}(d)$ and (ii) *additive* logit biases (Additive GRAPE) arising from unipotent actions in the general linear group GL. In Multiplicative GRAPE, a position $n \in \mathbb{Z}$ (or $t \in \mathbb{R}$) acts as $\mathbf{G}(n) = \exp(n\,\omega\,\mathbf{L})$ with a rank-2 skew generator $\mathbf{L} \in \mathbb{R}^{d \times d}$, yielding a relative, compositional, norm-preserving map with a closed-form matrix exponential. RoPE is recovered exactly when the $d/2$ planes are the canonical coordinate pairs with log-uniform spectrum. Learned commuting subspaces and compact non-commuting mixtures strictly extend this geometry to capture cross-subspace feature coupling at $O(d)$ and $O(rd)$ cost per head, respectively. In Additive GRAPE, additive logits arise as rank-1 (or low-rank) unipotent actions, recovering ALiBi and the Forgetting Transformer (FoX) as exact special cases while preserving an exact relative law and streaming cacheability. Altogether, GRAPE supplies a principled design space for positional geometry in long-context models, subsuming RoPE and ALiBi as special cases. Project Page: https://github.com/model-architectures/GRAPE.

## 1 INTRODUCTION

Positional information is essential for sequence modeling with Transformers (Vaswani et al., 2017), whose self-attention is otherwise permutation-invariant. Early work injected absolute positional codes (sinusoidal or learned) into token representations (Vaswani et al., 2017). Later, relative encodings depending on offsets (Shaw et al., 2018) and linear logit biases such as ALiBi (Press et al., 2021) were introduced, the latter offering strong length extrapolation with negligible overhead.

Rotary Position Embedding (RoPE) (Su et al., 2024) realizes relative positions as orthogonal planar rotations of queries and keys, preserving norms and yielding exact origin invariance of attention scores. Despite its appeal, RoPE fixes coordinate planes and typically a log-uniform spectrum, limiting cross-subspace coupling and contextual warping of phase. More broadly, absolute codes break translation equivariance; table-based relatives add window-dependent overhead. A new formulation is needed because current methods isolate the essential properties of stability, monotonic distance penalty, and expressivity. These observations motivate a unified formulation that (i) preserves RoPE's orthogonality and exact relativity when desired, (ii) *also* covers additive/forgetting mechanisms such as ALiBi (Press et al., 2021) and Forgetting Transformer (FoX) (Lin et al., 2025), and (iii) admits learned and contextual generalizations with clean streaming.

We therefore propose **G**roup **R**epresent**A**tional **P**osition **E**ncoding (**GRAPE**), a group-theoretic framework that unifies two complementary families of positional mechanisms (see Figure 1 for an overview). The multiplicative family (Multiplicative GRAPE) models positions as norm-preserving rotations in $\mathrm{SO}(d)$ acting on $(\mathbf{q}, \mathbf{k})$; the additive family (Additive GRAPE/Path-Integral Additive GRAPE) models positions as unipotent actions in the general linear group GL that yield

---

[*]Core contribution; [†]Corresponding authors.

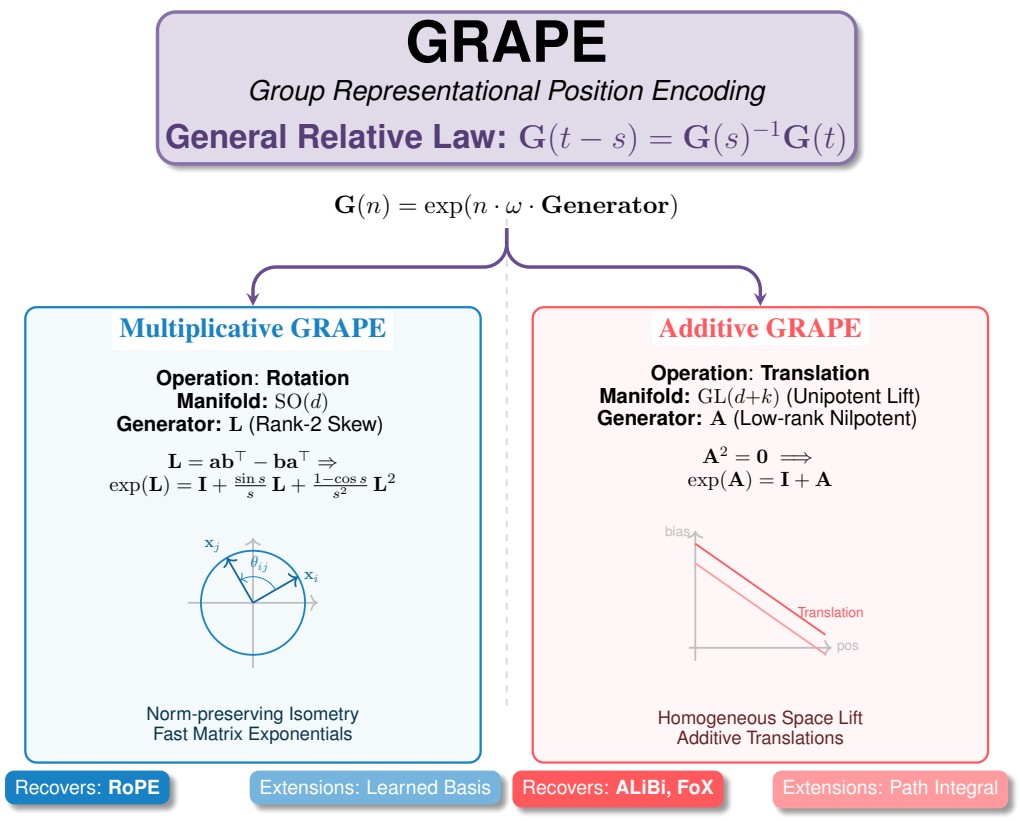

Figure 1: **Overview of the GRAPE Framework.** We unify positional encodings via group actions $\mathbf{G}(n) = \exp(n\omega\mathbf{L})$. **Left:** Multiplicative GRAPE recovers RoPE via rank-2 skew generators in $\mathrm{SO}(d)$. **Right:** Additive GRAPE recovers ALiBi and FoX via low-rank nilpotent generators in the unipotent subgroup of $\mathrm{GL}(d+k)$ ($k = 1$ or $2$).

linear-in-offset logit biases (including content-gated and path-integral forms). This perspective recovers RoPE and ALiBi as exact special cases, proves that FoX is an exact instance of Additive GRAPE, and supplies principled, streaming-friendly contextual extensions on both sides.

Concretely: *(a)* Multiplicative GRAPE (GRAPE-M) encodes $n \in \mathbb{Z}$ (or $t \in \mathbb{R}$) as an element of $\mathrm{SO}(d)$ via a rank-2 skew generator; and *(b)* Additive GRAPE (GRAPE-A) and Path-Integral Additive GRAPE (GRAPE-AP) lifts to the general linear group GL using homogeneous coordinates to produce linear-in-offset logit biases (recovering ALiBi and FoX).

For Multiplicative GRAPE, positions are mapped as

$$\mathbf{G}(n) = \exp\left(n\,\omega\,\mathbf{L}\right) \in \mathrm{SO}(d), \qquad \mathbf{L} = \mathbf{a}\mathbf{b}^\top - \mathbf{b}\mathbf{a}^\top \in \mathfrak{so}(d),$$

where $\mathbf{a}, \mathbf{b} \in \mathbb{R}^d$ define a rank-2 skew generator $\mathbf{L}$ and $\omega > 0$ is a frequency. The action is an isometry, and $\mathbf{G}(n+m) = \mathbf{G}(n)\mathbf{G}(m)$ guarantees exact origin invariance of attention logits. We derive a closed-form Rodrigues-type formula (Rodrigues, 1840; Hall, 2013), enabling fast linear-time application with stable derivatives and no explicit matrix materialization. RoPE is recovered when $d/2$ commuting rank-2 generators act on disjoint coordinate planes with prescribed frequencies.

For Additive GRAPE, positions are mapped via the matrix exponential $\mathbf{G}_{\mathrm{add}}(n) = \exp(n\omega\mathbf{A}) = \mathbf{I} + n\omega\mathbf{A}$ in a lifted homogeneous space. Here, the generator $\mathbf{A} \in \mathfrak{gl}(d+1)$ is a nilpotent matrix of rank one. While this additive transformation is not an isometry, it preserves the exact relative law, ensuring attention scores depend only on position offsets. This formulation provides a rigorous group-theoretic foundation for additive biases, recovering ALiBi and FoX as exact instances.

Our contributions are highlighted as follows:

1. We propose **GRAPE** as a unified group-theoretic view that subsumes *multiplicative* orthogonal rotations in $\mathrm{SO}(d)$ and *additive* unipotent (all eigenvalues equal to 1) mechanisms in general linear group GL, recovering RoPE and ALiBi as exact special cases and proving FoX is an exact instance (Appendix C).

2. **Multiplicative GRAPE.** We derive a closed-form rank-2 matrix exponential with fast application and stable differentiation; we show RoPE is a special multiplicative GRAPE in a possibly learned orthogonal basis.

3. **Additive GRAPE.** We show that linear-in-offset logit biases arise from rank-1 (or low-rank) unipotent actions in the general linear group GL with an exact relative law and streaming cacheability. This includes query- or key-gated slopes, a commuting dictionary of additive components, and exact recoveries of ALiBi and FoX in closed form (Sections 4, 4.2, Appendix C). We also formalize path-integral additive biases that remain causal and support efficient training. (Section 5).

## 2  MULTIPLICATIVE GROUP REPRESENTATIONAL POSITION ENCODING

We propose the **Multiplicative GRAPE**, as a Lie-group positional map with a closed-form rank-2 matrix exponential, an exact relative law, and a streaming/cache methodology. The core intuition is to encode position as a norm-preserving rotation in the special orthogonal group $\mathrm{SO}(d)$ [1](Hall, 2013). A single skew-symmetric generator $\mathbf{L} \in \mathfrak{so}(d)$ produces the entire family of rotations via the matrix exponential. We begin with notation and the rank-2 generator.

### 2.1  PRELIMINARIES AND RANK-2 GENERATOR

The generator $\mathbf{L}$ is formally defined as an element of the corresponding Lie algebra, $\mathfrak{so}(d)$. Let $\mathfrak{so}(d) = \{\mathbf{L} \in \mathbb{R}^{d \times d} : \mathbf{L}^\top = -\mathbf{L}\}$ denote the Lie algebra of $\mathrm{SO}(d)$. The simplest non-trivial generator defines a rotation within a single 2D plane. We construct such a rank-2 generator from two vectors, $\mathbf{a}$ and $\mathbf{b}$, that span this plane of action. For $\mathbf{a}, \mathbf{b} \in \mathbb{R}^d$, define the rank-2 generator $\mathbf{L} \equiv \mathbf{L}(\mathbf{a}, \mathbf{b})$ as

$$\mathbf{L}(\mathbf{a}, \mathbf{b}) = \mathbf{a}\mathbf{b}^\top - \mathbf{b}\mathbf{a}^\top, \; \alpha = \|\mathbf{a}\|^2, \; \beta = \|\mathbf{b}\|^2, \; \gamma = \mathbf{a}^\top \mathbf{b}, \Delta = \alpha\beta - \gamma^2 \geq 0, \; s = \sqrt{\Delta}. \quad (2.1)$$

**Rank-2 structure.** Let $\mathcal{U} = \mathrm{span}\{\mathbf{a}, \mathbf{b}\}$. The rank-2 generator $\mathbf{L}$ has a useful geometric property: applying it twice projects onto the action plane $\mathcal{U}$ and scales. A direct calculation shows

$$\mathbf{L}^2 = -s^2 \, \mathbf{P}_{\mathcal{U}},$$

where $\mathbf{P}_{\mathcal{U}}$ is the orthogonal projector to the space $\mathcal{U}$. Hence spectrum of $\mathbf{L}$ (the set of its eigenvalues), denoted $\sigma(\mathbf{L})$, is $\{\pm is, 0, \ldots, 0\}$ and the minimal polynomial is $\lambda(\lambda^2 + s^2)$. A detailed derivation is given in Appendix I.

**Initialization.** Write $\mathbf{A} \triangleq [\mathbf{a} \; \mathbf{b}] \in \mathbb{R}^{d \times 2}$ and $\mathbf{J} = \left(\begin{smallmatrix} 0 & -1 \\ 1 & 0 \end{smallmatrix}\right)$ so that $\mathbf{L} = \mathbf{A}\mathbf{J}\mathbf{A}^\top$. For any $\mathbf{M} \in \mathrm{SL}(2)$ (the $2 \times 2$ real matrices with determinant 1, see Table 3), $\mathbf{M}\mathbf{J}\mathbf{M}^\top = \mathbf{J}$ and thus $\mathbf{A} \mapsto \mathbf{A}\mathbf{M}$ leaves $\mathbf{L}$ invariant; for general $\mathbf{M} \in \mathrm{GL}(2)$ the group of invertible $2 \times 2$ matrices), $\mathbf{L}$ scales by $\det(\mathbf{M})$. Therefore the oriented plane $\mathcal{U} = \mathrm{span}\{\mathbf{a}, \mathbf{b}\}$ and the scalar $s = \sqrt{\alpha\beta - \gamma^2}$ determine the action. We fix a gauge at initialization by $\|\mathbf{a}\| = \|\mathbf{b}\| = 1$ and $\mathbf{a}^\top \mathbf{b} = 0$ (absorbing scale into $\omega$).

**Canonical $90°$ rotation operator.** Fix a block-diagonal complex structure $\mathcal{J} \in \mathfrak{so}(d)$ with $\mathcal{J}^\top = -\mathcal{J}$ and $\mathcal{J}^2 = -\mathbf{I}$ (for odd $d$, act on the top-left $2\lfloor d/2 \rfloor$ coordinates and leave the final coordinate unchanged). Concretely, $\mathcal{J} = \bigoplus_{i=1}^{\lfloor d/2 \rfloor} \left(\begin{smallmatrix} 0 & -1 \\ 1 & 0 \end{smallmatrix}\right)$. For any $\mathbf{a} \in \mathbb{R}^d$, write $\mathbf{a}_\perp := \mathcal{J}\mathbf{a}$, which equals "$\mathbf{a}$ rotated by $90°$" within the canonical 2D blocks and satisfies $\mathbf{a}^\top \mathbf{a}_\perp = 0$ and $\|\mathbf{a}_\perp\| = \|\mathbf{a}\|$.

### 2.2  EXACT RELATIVE LAW

For a fixed $\mathbf{L} \in \mathfrak{so}(d)$, define $\mathbf{G}(n) = \exp(n\mathbf{L}) \in \mathrm{SO}(d)$, which forms a one-parameter subgroup. The exact relative law property for positional encoding implies:

$$\mathbf{G}(t-s) = \mathbf{G}(s)^\top \mathbf{G}(t), \qquad \mathbf{G}(n)^\top \mathbf{G}(n) = \mathbf{I}.$$

---

[1]Definitions of $\mathrm{SO}(d)$ and other mathematical terms are postponed to Table 3 in the Appendix.

Here $\mathbf{G}(n) \in \mathrm{SO}(d)$, so the transpose coincides with the group inverse, $\mathbf{G}(n)^\top = \mathbf{G}(n)^{-1}$; the identity above is exactly the **relative-position law for a one-parameter subgroup**. A concise summary of $\mathrm{SO}(d)$, $\mathrm{GL}(d)$ and $\mathrm{SL}(d)$ is collected in Table 3. This algebraic property enables relative positional encoding: interactions depend only on offsets.

$$\mathbf{G}(n) = \exp(n\omega\mathbf{L}), \quad \mathbf{G}(n+m) = \mathbf{G}(n)\mathbf{G}(m), \quad \mathbf{G}(0) = \mathbf{I}, \quad \text{and} \quad \mathbf{G}(-n) = \mathbf{G}(n)^\top.$$

Crucially, this exact relative property relies solely on the one-parameter subgroup structure ($G(n+m) = G(n)G(m)$), holding true regardless of whether the generator implies commuting or coupled non-commuting subspaces.

## 2.3 CLOSED-FORM FAST MATRIX EXPONENTIAL

Based on the minimal polynomial mentioned in Section 2.1, the exponential map $\exp(\mathbf{L})$ for a rank-2 generator can be expressed as a quadratic in $\mathbf{L}$. This yields a convenient closed-form solution, often referred to as a Rodrigues-type formula (Rodrigues, 1840; Hall, 2013):

$$\exp(\mathbf{L}) = \mathbf{I} + \frac{\sin s}{s}\mathbf{L} + \frac{1 - \cos s}{s^2}\mathbf{L}^2.$$

Geometrically, the formula is best understood via $\mathbf{L}^2$ as a projector onto $\mathcal{U}$. Since $\mathbf{L}^2 = -s^2\mathbf{P}_\mathcal{U}$, the exponential can be written as

$$\exp(\mathbf{L}) = \mathbf{I} - (1 - \cos s)\,\mathbf{P}_\mathcal{U} + \frac{\sin s}{s}\mathbf{L},$$

which reveals its action explicitly: it is a rotation by angle $s$ within the plane $\mathcal{U} = \mathrm{span}\{\mathbf{a}, \mathbf{b}\}$ and the identity on the orthogonal complement $\mathcal{U}^\perp$. The vectors $\mathbf{a}$ and $\mathbf{b}$ thus define the plane of action for the positional rotation.

**Cost of application.** For a single rank-2 plane, computing $\mathbf{y} = \mathbf{G}(n)\mathbf{x}$ requires two inner products $u = \langle \mathbf{a}, \mathbf{x}\rangle$, $v = \langle \mathbf{b}, \mathbf{x}\rangle$, followed by $\mathbf{y} = \mathbf{x} + f_1(n)(\mathbf{a}v - \mathbf{b}u) + f_2(n)\left[\gamma(\mathbf{a}v + \mathbf{b}u) - \beta\mathbf{a}u - \alpha\mathbf{b}v\right]$, where $(\alpha, \beta, \gamma)$ are plane scalars and $f_{1,2}$ are trigonometric scalars (with series guards as $s \to 0$). This is $O(d)$ flops with a small constant and no materialization of $\mathbf{G}(n)$; derivative expressions are in Appendix I.

## 2.4 THE $\mathbf{b} = \mathcal{J}\mathbf{a}$ CONSTRAINT

We now consider an important special case by setting $\mathbf{b} = \mathcal{J}\mathbf{a}$. This constraint, which makes the plane vectors $\mathbf{a}$ and $\mathbf{b}$ orthogonal and equal in norm, significantly simplifies the generator's structure and reveals a direct connection to the canonical RoPE formulation. With this constraint, the scalars simplify: $\gamma = \mathbf{a}^\top\mathbf{b} = \mathbf{a}^\top\mathcal{J}\mathbf{a} = 0$, $\beta = \|\mathbf{b}\|^2 = \|\mathbf{a}\|^2 = \alpha$, and hence $s = \sqrt{\alpha\beta - \gamma^2} = \alpha$. Moreover, on the 2D subspace $\mathcal{U} = \mathrm{span}\{\mathbf{a}, \mathcal{J}\mathbf{a}\}$ one has

$$\mathbf{L}(\mathbf{a}, \mathcal{J}\mathbf{a})\mathbf{a} = -(\mathcal{J}\mathbf{a})\alpha, \qquad \mathbf{L}(\mathbf{a}, \mathcal{J}\mathbf{a})\,\mathcal{J}\mathbf{a} = \alpha\,\mathbf{a},$$

so $\mathbf{L}(\mathbf{a}, \mathcal{J}\mathbf{a})|_\mathcal{U} = -\alpha\,\mathcal{J}|_\mathcal{U}$ and $\mathbf{L}(\mathbf{a}, \mathcal{J}\mathbf{a})|_{\mathcal{U}^\perp} = 0$. Therefore

$$\exp\left(n\omega\mathbf{L}(\mathbf{a}, \mathcal{J}\mathbf{a})\right) = \mathbf{I} - \left(1 - \cos(n\omega\alpha)\right)\mathbf{P}_\mathcal{U} - \sin(n\omega\alpha)\,\mathcal{J}\mathbf{P}_\mathcal{U},$$

This expression follows by substituting $\mathbf{L}|_\mathcal{U} = -\alpha\,\mathcal{J}|_\mathcal{U}$ and $\mathbf{L}^2 = -\alpha^2\mathbf{P}_\mathcal{U}$ into the Rodrigues formula $\exp(n\omega\mathbf{L}) = \mathbf{I} + \frac{\sin(n\omega s)}{s}\mathbf{L} + \frac{1 - \cos(n\omega s)}{s^2}\mathbf{L}^2$ with $s = \alpha$; see Appendix I for the algebraic steps. It is a pure planar rotation by angle $n\omega\alpha$ on $\mathcal{U}$ and the identity on $\mathcal{U}^\perp$.

**Corollary 2.1** (Frequency–norm coupling). If $\|\mathbf{a}\| = 1$, the rotation angle reduces to $n\omega$. Without normalization, the effective frequency is $\omega_{\mathrm{eff}} = \omega\|\mathbf{a}\|^2$, so the scale of $a$ can be absorbed into $\omega$.

## 2.5 APPLICATION TO RELATIVE ENCODING AND EQUIVARIANCE

We now demonstrate how the **GRAPE-M** operator $\mathbf{G}(n)$ is applied in practice. As established in Section 2.2, the operator's group structure guarantees the exact relative law. We first transform the query and key vectors, $\mathbf{q}_i$ and $\mathbf{k}_j$, into position-aware representations, $\widetilde{\mathbf{q}}_i$ and $\widetilde{\mathbf{k}}_j$:

$$\widetilde{\mathbf{q}}_i := \mathbf{G}(i)\mathbf{q}_i, \qquad \widetilde{\mathbf{k}}_j := \mathbf{G}(j)\mathbf{k}_j.$$

It follows from the exact relative law established in Section 2.2 that the attention score between these position-aware vectors simplifies to:

$$\widetilde{\mathbf{q}}_i^\top \widetilde{\mathbf{k}}_j = \mathbf{q}_i^\top \mathbf{G}(i)^\top \mathbf{G}(j)\mathbf{k}_j = \mathbf{q}_i^\top \mathbf{G}(j-i)\mathbf{k}_j.$$

Hence, the attention score depends solely on the relative offset $j - i$, not on the absolute positions.

**Streaming and caching.** At inference, cache $\mathbf{k}_j^\star = \mathbf{G}(j)\mathbf{k}_j$ once when token $j$ arrives. At step $t$, form $\widetilde{\mathbf{q}}_t = \mathbf{G}(t)\mathbf{q}_t$ and compute logits $\widetilde{\mathbf{q}}_t^\top \mathbf{k}_j^\star$. No cache rotation is needed when $t$ increments; complexity matches RoPE. A full integration into multi-head attention (per-head formulation, logits, and streaming) is detailed in Section B.

## 3 MULTI-SUBSPACE MULTIPLICATIVE GRAPE

A single rank-2 generator acts on a 2D subspace, leaving the rest of the $d$-dimensional space untouched. To encode position across the entire hidden dimension, we can combine multiple generators. This leads to the Multi-Subspace (MS) **Multiplicative GRAPE (GRAPE-M)** model, which forms the basis for both RoPE and more expressive types. Detailed rank-2 algebra appears in Appendix I.

### 3.1 MULTI-SUBSPACE GRAPE-M AND RoPE AS A SPECIAL CASE

The simplest way to combine generators is to ensure they act on mutually orthogonal subspaces, which guarantees they commute. Let $d$ be even. For $i = 1, \ldots, d/2$, we can define a set of rank-2 generators $\{\mathbf{L}_i\}$, each acting on a distinct 2D plane. RoPE is the canonical example of this construction. We further discussed non-commuting multiplicative GRAPE in Appendix D.

Let the $2 \times 2$ canonical skew matrix be $\mathbf{J} = \left( \begin{smallmatrix} 0 & -1 \\ 1 & 0 \end{smallmatrix} \right)$ and the coordinate selector be $\mathbf{U}_i = [\mathbf{e}_{2i-1}\ \mathbf{e}_{2i}] \in \mathbb{R}^{d \times 2}$. We set the rank-2 generators as $\mathbf{L}_i = \mathbf{U}_i \mathbf{J} \mathbf{U}_i^\top = \mathbf{L}(\mathbf{e}_{2i-1}, \mathbf{e}_{2i})$ and assign per-plane frequencies $\theta_i > 0$. The total generator is the commuting sum:

$$\mathbf{L}_{\text{RoPE}} = \sum_{i=1}^{d/2} \theta_i \mathbf{L}_i \qquad \text{with} \qquad [\mathbf{L}_i, \mathbf{L}_j] = 0 \text{ for } i \neq j.$$

Then

$$\mathbf{G}(n) = \exp\left(n\mathbf{L}_{\text{RoPE}}\right) = \prod_{i=1}^{d/2} \exp(n\theta_i \mathbf{L}_i) = \text{blockdiag}\left(\mathbf{R}_2(n\theta_1), \ldots, \mathbf{R}_2(n\theta_{d/2})\right), \quad (3.1)$$

where $\mathbf{R}_2(\theta)$ denotes the standard $2 \times 2$ rotation matrix introduced in Table 3, and the last equality holds because each term $\exp(n\theta_i \mathbf{L}_i)$ is identity except for a single $2\times 2$ rotation block on its diagonal. Eq. (3.1) is precisely the RoPE mapping: a block-diagonal product of planar rotations with per-subspace angles $n\theta_i$.

Equality holds when the planes $\{\mathbf{U}_i\}$ are the coordinate 2D blocks and $\{\theta_i\}$ follow the canonical log-uniform spectrum.

**Proposition 3.1** (RoPE is a multiplicative GRAPE). Choose $d/2$ mutually orthogonal vectors $\{\mathbf{a}_i\}$ and set $\mathbf{b}_i = \mathcal{J}\mathbf{a}_i$ with per-plane angles $\theta_i$. Then the commuting MS-GRAPE $\mathbf{G}(n) = \prod_{i=1}^{d/2} \exp(n\theta_i \mathbf{L}(\mathbf{a}_i, \mathcal{J}\mathbf{a}_i))$ equals the standard RoPE map in a (possibly learned) orthogonal basis. If the planes are the canonical coordinate pairs and $\{\theta_i\}$ follow the log-uniform spectrum, we recover the canonical RoPE exactly.

**Spectral parameterization.** Classical RoPE chooses $\theta_i$ on a log-uniform grid across $i$. In GRAPE, $\theta_i$ can be learned or shared/tied across heads or layers. The MS-GRAPE view also allows replacing the coordinate selectors $\mathbf{U}_i$ by a learned orthogonal basis $\mathbf{B} \in \text{SO}(d)$ so that $\mathbf{L} = \sum_i \theta_i \mathbf{B}\mathbf{U}_i \mathbf{J} \mathbf{U}_i^\top \mathbf{B}^\top$, preserving commutativity while learning subspaces.

**Multimodal GRAPE.** Please refer to Appendix G for 2D and 3D GRAPE for Vision and Multimodal Position Encoding.

## 4 ADDITIVE GROUP REPRESENTATIONAL POSITION ENCODING

This section shows that additive positional mechanisms (absolute shifts of features and additive logit biases, including ALiBi (Press et al., 2021)) also admit a group-theoretic formulation. The key is a homogeneous lift to an augmented space and a one-parameter subgroup of the general linear group GL that acts by unipotent (all eigenvalues equal to 1) transformations. This yields an exact relative law and streaming/cache rules analogous to Section 2.5.

### 4.1 HOMOGENEOUS LIFT AND A UNIPOTENT ACTION

To produce additive biases from a multiplicative group action, we employ the homogeneous lift. This is a standard method in linear algebra for representing affine transformations (such as translations) as linear transformations in a higher-dimensional space. Let $\widehat{\mathbf{x}} \in \mathbb{R}^{d+k}$ denote a homogeneous augmentation of $\mathbf{x} \in \mathbb{R}^d$. We work within the general linear group $\mathrm{GL}(d+k)$ and its Lie algebra $\mathfrak{gl}(d+k)$. Fix a nilpotent generator $\mathbf{A}$ such that $\mathbf{A}^2 = \mathbf{0}$. Its exponential is unipotent:

$$\mathbf{G}_{\mathrm{add}}(n) := \exp(n\,\omega\,\mathbf{A}) = \mathbf{I} + n\,\omega\,\mathbf{A} \in \mathrm{GL}(d+k).$$

Crucially, this action forms a one-parameter subgroup: $\mathbf{G}_{\mathrm{add}}(n+m) = \mathbf{G}_{\mathrm{add}}(n)\mathbf{G}_{\mathrm{add}}(m)$.

**Canonical rank-1 generator in a** $(d+1)$**-lift.** A convenient concrete choice (used in Appendix J.3) is the rank-1 index-2 nilpotent

$$\mathbf{A} = \begin{bmatrix} \mathbf{0}_{d\times d} & \mathbf{u}_{\mathrm{shift}} \\ \mathbf{0}^\top & 0 \end{bmatrix} \in \mathbb{R}^{(d+1)\times(d+1)}, \qquad \mathbf{A}^2 = \mathbf{0}, \tag{4.1}$$

where $\mathbf{u}_{\mathrm{shift}} \in \mathbb{R}^d$; acting on $\widehat{\mathbf{x}} = [\mathbf{x}; 1]$ yields $\mathbf{G}_{\mathrm{add}}(n)\widehat{\mathbf{x}} = [\mathbf{x} + n\omega\mathbf{u}_{\mathrm{shift}}; 1]$, i.e., a translation in the original $d$-dimensional space.

**Application and exact relative law in** GL**.** For queries/keys augmented as $\widehat{\mathbf{q}}_i$ and $\widehat{\mathbf{k}}_j$, define

$$\widetilde{\mathbf{q}}_i := \mathbf{G}_{\mathrm{add}}(i)\,\widehat{\mathbf{q}}_i, \qquad \widetilde{\mathbf{k}}_j := \mathbf{G}_{\mathrm{add}}(j)^{-\top}\,\widehat{\mathbf{k}}_j. \tag{4.2}$$

We use the shorthand $\mathbf{G}_{\mathrm{add}}(j)^{-\top} := (\mathbf{G}_{\mathrm{add}}(j)^{-1})^\top$ to emphasize that we first take the group inverse in GL and then transpose it. This composition results in the final form:

$$\widetilde{\mathbf{q}}_i^\top \widetilde{\mathbf{k}}_j = \widehat{\mathbf{q}}_i^\top \mathbf{G}_{\mathrm{add}}(j-i)^{-\top} \widehat{\mathbf{k}}_j, \quad \text{depending only on } j-i. \tag{4.3}$$

**Closed form and content-gated additive term.** To incorporate content dependence while strictly maintaining the unipotent group structure, we decompose the action into separate commuting generators for queries and keys. Let $\mathbf{A}_0$ be the canonical rank-1 nilpotent basis in a $(d+2)$-dimensional lift (augmenting $\widehat{\mathbf{q}} = [\mathbf{q}; 1; 0]$ and $\widehat{\mathbf{k}} = [\mathbf{k}; 0; 1]$):

$$\mathbf{A}_0 = \mathbf{e}_{d+2}\mathbf{e}_{d+1}^\top \in \mathbb{R}^{(d+2)\times(d+2)}. \tag{4.4}$$

We modulate this basis via non-negative scalar gates $\lambda_q(\mathbf{q}_i) = \mathrm{softplus}(\mathbf{v}^\top \mathbf{q}_i/\sqrt{d})$ and $\lambda_k(\mathbf{k}_j) = \mathrm{softplus}(\mathbf{u}^\top \mathbf{k}_j/\sqrt{d})$.

Because the scaled generators share the same nilpotent basis, they commute. Let $\Lambda_{ij} := \lambda_q(\mathbf{q}_i) + \lambda_k(\mathbf{k}_j) \geq 0$ and define the effective (endpoint-dependent) nilpotent generator $\mathbf{A}_{ij} := -\Lambda_{ij}\,\mathbf{A}_0$, so that $\mathbf{A}_{ij}^\top = -\Lambda_{ij}\,\mathbf{A}_0^\top$. Their composition in the relative offset $m = j - i$ yields a summed decay rate. The unipotent inverse transpose admits an exact closed form that converts the offset $m$ (negative in causal attention) into a penalty:

$$\mathbf{G}_{\mathrm{add}}(m)^{-\top} = \mathbf{I} - m\,\omega\,\mathbf{A}_{ij}^\top = \mathbf{I} + m\,\omega\,(\lambda_q(\mathbf{q}_i) + \lambda_k(\mathbf{k}_j))\,\mathbf{A}_0^\top, \qquad m = j-i. \tag{4.5}$$

This formulation is rigorous: the $\mathrm{softplus}(\cdot)$ function ensures the effective decay rate is strictly non-negative. When $j \leq i$, the offset $m$ is negative, resulting in a monotonic penalty. Applying this operator yields the exact affine score:

$$\widetilde{\mathbf{q}}_i^\top \widetilde{\mathbf{k}}_j = \mathbf{q}_i^\top \mathbf{k}_j + (j-i)\,\omega\,[\mathrm{softplus}(\mathbf{v}^\top \mathbf{q}_i/\sqrt{d}) + \mathrm{softplus}(\mathbf{u}^\top \mathbf{k}_j/\sqrt{d})]. \tag{4.6}$$

Note that under the asymmetric lift $\widehat{\mathbf{q}} = [\mathbf{q}; 1; 0]$, $\widehat{\mathbf{k}} = [\mathbf{k}; 0; 1]$, the base inner product satisfies $\widehat{\mathbf{q}}_i^\top \widehat{\mathbf{k}}_j = \mathbf{q}_i^\top \mathbf{k}_j$ (no extra constant), while the rank-1 update contributes exactly $(j-i)\omega(\lambda_q + \lambda_k)$, which is non-positive for causal offsets $j \leq i$, which we denoted as **GRAPE-A-QK**. This derives a learnable, content-adaptive linear bias (slope) from first principles: it is the sum of query-dependent and key-dependent unipotent actions in the general linear group.

## 4.2 EXACT ALiBi AS A RANK-1 UNIPOTENT IN $\mathrm{GL}(d+2)$

ALiBi adds a head-specific scalar slope $\beta_h(j-i)$ to the logits that is independent of content. This is captured exactly by augmenting with two constant coordinates:

$$\widehat{\mathbf{q}}_i = [\mathbf{q}_i;\ 1;\ 0] \in \mathbb{R}^{d+2}, \qquad \widehat{\mathbf{k}}_j = [\mathbf{k}_j;\ 0;\ 1] \in \mathbb{R}^{d+2},$$

and choosing the rank-1 nilpotent generator

$$\mathbf{A}_h^\top = -\beta_h\,\mathbf{e}_{d+1}\,\mathbf{e}_{d+2}^\top \quad \Longleftrightarrow \quad \mathbf{A}_h = -\beta_h\,\mathbf{e}_{d+2}\,\mathbf{e}_{d+1}^\top, \qquad (\mathbf{A}_h^\top)^2 = \mathbf{0}. \tag{4.7}$$

Then $\mathbf{G}_{\mathrm{add},h}(m)^{-\top} = \mathbf{I} - m\,\mathbf{A}_h^\top$ and

$$\widehat{\mathbf{q}}_i^\top\,\mathbf{G}_{\mathrm{add},h}(j-i)^{-\top}\,\widehat{\mathbf{k}}_j = \mathbf{q}_i^\top \mathbf{k}_j\ +\ (j-i)\,\beta_h,$$

i.e., the ALiBi term emerges as a unipotent $\mathrm{GL}(d+2)$ action with exact relative composition.

**Remark 4.1** (Why $\mathrm{GL}(d+2)$). While general additive biases, such as key-gated slopes, can be realized in $\mathrm{GL}(d+1)$, the content-independent nature of exact ALiBi necessitates the $(d+2)$-dimensional lift. Generating a non-zero scalar bias via a nilpotent generator requires the interaction of two distinct constant coordinates, one for query, one for key, to avoid self-loops that would violate the trace-zero property of the generator.

**FoX as GRAPE-A.** Let $f_t \in (0,1]$ be per-token forget scalars and set $\omega_t := \log f_t$. The resulting additive bias is $b(t,j) = \sum_{\ell=j+1}^{t} \omega_\ell$, which coincides with FoX's forgetting bias $D_{ij}$. A full derivation and the unipotent path product are given in Appendix C.

## 5 PATH INTEGRAL ADDITIVE GRAPE

Additive GRAPE (GRAPE-A) realizes exactly relative additive logits via a one-parameter unipotent action in the general linear group $\mathrm{GL}$; the bias depends only on an offset $m = j-i$ (or a contextual phase difference $\Phi_j - \Phi_i$ when using cumulative phases). Here the "phase" $\Phi_t$ is a scalar path variable, typically defined as a cumulative sum $\Phi_t = \sum_{\ell < t} \omega_\ell$ of per-token frequencies $\omega_\ell$, so that $\Phi_j - \Phi_i$ plays the role of an effective relative position. In practice, we sometimes want the amount of additive encouragement/suppression between a key at $j$ and a query at $t$ to depend on the endpoint $t$ (e.g., the current syntactic or semantic needs of the query token), while preserving causality, boundedness, and clean composition with the orthogonal GRAPE acting on $(\mathbf{q}, \mathbf{k})$. We formalize this by a rigorously defined path-integral sum, deriving conditions under which the exact relative law of Additive GRAPE is recovered.

**Definition (Path-integral bias).** Fix a head $h$ and per-head scale $\alpha_h > 0$. For each time $u$, let $\mathbf{p}_{u,h} \in \mathbb{R}^d$ be a positional embedding obtained from token-local features (a linear projection followed by RMS normalization in our implementation). Let $\mathcal{J}$ be the canonical block-diagonal $90°$ operator (Section 2.4), and define $\mathbf{R}_\ell := \exp(\ell\,\mathcal{J})$ (a fixed commuting rotation). For a link function $g : \mathbb{R} \to (-\infty, 0)$ that is monotone increasing and 1-Lipschitz[2], define the *edge potential*

$$\psi_h(t,\ell) := \alpha_h\, g\left(\frac{1}{d}\left\langle \mathbf{p}_{t,h}, \mathbf{R}_\ell\,\mathbf{p}_{\ell,h}\right\rangle\right) \leq 0, \qquad \ell < t. \tag{5.1}$$

The vectors $\mathbf{p}_{t,h}$ and $\mathbf{p}_{\ell,h}$ here are the positional embedding. The *path-integral additive bias* from key position $j$ to query position $t$ is the causal sum

$$b_h(t,j) := \sum_{\ell=j+1}^{t} \psi_h(t,\ell) \ \leq\ 0. \tag{5.2}$$

The attention logit combines this additive term with either the raw or orthogonally-rotary bilinear part:

$$\ell_{t,j,h} = \frac{1}{\sqrt{d}}\,\mathbf{q}_{t,h}^\top \mathbf{k}_{j,h} + b_h(t,j) \quad \text{or} \quad \ell_{t,j,h} = \frac{1}{\sqrt{d}}\,\mathbf{q}_{t,h}^\top \mathbf{G}_h(j-t)\mathbf{k}_{j,h} + b_h(t,j). \tag{5.3}$$

---

[2]Our experiments take $g(z) = \log(\mathrm{Sigmoid}(z))$; then $g'(z) = 1 - \mathrm{Sigmoid}(z) \in (0,1)$, ensuring 1-Lipschitzness.

**Group-theoretic formalization and path composition.** Let $\mathbf{E} \in \mathbb{R}^{(d+2)\times(d+2)}$ be a fixed rank-1 nilpotent with $\mathbf{E}^2 = \mathbf{0}$ (e.g., $\mathbf{E} = \mathbf{e}_{d+2}\mathbf{e}_{d+1}^\top$ as in Section 4.2). For each fixed endpoint $t$, define endpoint-indexed unipotent factors

$$\mathbf{H}_h^{(t)}(\ell) := \mathbf{I} - \psi_h(t,\ell)\,\mathbf{E}.$$

Since $\mathbf{E}^2 = 0$, the path product along $(j, t]$ collapses additively:

$$\prod_{\ell=j+1}^{t} \mathbf{H}_h^{(t)}(\ell) = \mathbf{I} - \left( \sum_{\ell=j+1}^{t} \psi_h(t,\ell) \right) \mathbf{E} = \mathbf{I} - b_h(t,j)\,\mathbf{E}. \tag{5.4}$$

Scoring in homogeneous coordinates as in Section 4 with the paired inverse-transpose removes multiplicative anisotropy and yields exactly the additive term $b_h(t,j)$, cf. Eq. (4.3), since $\big(\mathbf{I} - b_h(t,j)\mathbf{E}\big)^{-\top} = \mathbf{I} + b_h(t,j)\mathbf{E}^\top$ and $\widehat{\mathbf{q}}_{t,h}^\top \mathbf{E}^\top \widehat{\mathbf{k}}_{j,h} = 1$ under the asymmetric lift. The *rowwise* semigroup law is preserved (Eq. (5.4)), while the $t$-dependence of the factors intentionally relaxes the global one-parameter group law.

**Relation to GRAPE-A.** GRAPE-AP strictly contains GRAPE-A as the special case in which edge potentials do not depend on the endpoint:

$$\psi_h(t,\ell) \equiv -\,\theta_h\, a_\ell, \qquad \theta_h \geq 0,\ a_\ell \geq 0 \implies$$

$$b_h(t,j) = -\theta_h \sum_{\ell=j+1}^{t} a_\ell = -\theta_h\big(A_t - A_j\big) \leq 0, \quad A_u := \sum_{\ell<u} a_\ell.$$

Two important instances follow directly:

- **Exact ALiBi.** $a_\ell \equiv 1$ gives $b_h(t,j) = -\theta_h(t-j) = \theta_h(j-t)$; setting $\theta_h = \beta_h$ recovers the ALiBi term from Section 4.2.
- **Phase-modulated Additive GRAPE.** If $a_\ell = \omega_\ell$ with $\omega_\ell = g(x_\ell) \geq 0$, then $b_h(t,j) = -\theta_h(\Phi_t - \Phi_j)$ with $\Phi_u = \sum_{\ell<u} \omega_\ell$.

In both cases, $b_h(t,j)$ depends only on a (possibly contextual) phase difference and thus obeys the exact relative law with the same streaming/cache policy as Section 4. Outside these endpoint-independent regimes, GRAPE-AP provides strictly more expressive, path-integral biases while preserving row-wise path composition (Eq. (5.4)).

**Computation and streaming.** For each head $h$ and decoding step $t$, compute the row $\{\psi_h(t,\ell)\}_{\ell\leq t}$ by a single similarity sweep $\ell \mapsto \langle \mathbf{p}_{t,h}, \mathbf{R}_\ell \mathbf{p}_{\ell,h} \rangle$ (the rotated probes $\mathbf{R}_\ell \mathbf{p}_{\ell,h}$ can be cached on arrival), apply the link $g$, and take a prefix sum to obtain $j \mapsto b_h(t,j)$. This yields $O(t)$ per-step overhead with $O(1)$ recomputation per cached key; memory is $O(L)$ per head for the cached probes (or $O(d)$ if the per-$\ell$ rotations are recomputed on the fly).

**Spectral and stability.** Each factor $\mathbf{H}_h^{(t)}(\ell) = \mathbf{I} - \psi_h(t,\ell)\mathbf{E}$ is unipotent with all eigenvalues 1 and at most two singular values deviating from 1; the full path product equals $\mathbf{I} - b_h(t,j)\mathbf{E}$ (Eq. (5.4)). As in Appendix J.3, the paired inverse-transpose used for scoring cancels multiplicative distortions and delivers exactly the additive bias $b_h(t,j)$; operator norms remain controlled linearly in $|b_h(t,j)|$.

A more extensive spectral analysis, including eigenvalue structure and singular-value behavior across GRAPE variants, is provided in Appendix J. There, we also give an explicit comparison to PaTH Attention (Yang et al., 2025b), which is shown to be contractive and near singular. These properties may impair PaTH's effectiveness in long-context modeling.

# 6 EXPERIMENTS

In this section, we evaluate the performance of GRAPE on the language modeling task in comparison with baseline positional encoding mechanisms, including RoPE (Su et al., 2024), AliBi (Press et al., 2021), as well as Forgetting Transformer (FoX) (Lin et al., 2025).

## 6.1 IMPLEMENTATION DETAILS

Based on the nanoGPT codebase (Karpathy, 2022), our experiments are implemented based on the Llama model (Touvron et al., 2023a). We only change the positional encoding mechanism

and keep the rest of the model architecture the same as Llama. We choose FineWeb-Edu 100B dataset (Lozhkov et al., 2024), which contains 100 billion training tokens and 0.1 billion validation tokens, and we randomly choose 50B tokens for training. Our medium models (353M) have 24 layers and 8 heads, with a hidden size of 1024; large models (770M) have 36 layers and 10 heads, with a hidden size of 1280 and a head dimension of 128. We applied QK RMSNorm for training stability (Yang et al., 2025a). The context length is set to 4,096, and the batch size is 480. All the large models (770M) are optimized by AdamW optimizer (Loshchilov and Hutter, 2019), with a maximum learning rate of $2 \times 10^{-3}$ for medium models and $1 \times 10^{-3}$ for large models, $(\beta_1, \beta_2) = 0.9, 0.95$, and a weight decay of 0.01. We use a cosine learning rate scheduler with 2,000 warm-up iterations, and the minimum learning rate is $3 \times 10^{-5}$. We also clip the gradient to 1.0 for stabler training. The frequency of RoPE is set to 10,000. Moreover, for fair comparison, we do not use FoX-Pro and disabled the KV-shift module within it.

## 6.2 RESULT ANALYSIS

The curves for training and validation loss of models with a variant positional encoding mechanism are displayed in Figures 2 and 3. This analysis provides specific insight into the source of the framework's stability and performance. It can be observed that GRAPE can keep a persistent edge over other mechanisms, including RoPE and FoX. Moreover, the model with RoPE suffers from training instability shown in Figure 3 (a), while the model with GRAPE embedding steadily improves during the training process.

Table 1: The evaluation results of medium models with different positional encoding mechanisms pre-trained using the FineWeb-Edu 100B dataset (0-shot with lm-evaluation-harness). Avg. is re-computed over the remaining 7 tasks. The best scores in each column are **bolded**, and the second best are underlined. Results are ranked separately for w/o KV-shift and w/ KV-shift.

| Method | ARC-E | ARC-C | HellaSwag | OBQA | PIQA | WinoGrande | SciQ | Avg. |
|---|---|---|---|---|---|---|---|---|
| RoPE | 56.36 | 30.38 | 44.65 | 35.20 | 68.77 | 52.33 | 74.40 | 51.73 |
| GRAPE-M-ctx | 56.31 | 29.95 | 44.37 | 33.40 | 68.23 | 53.28 | 76.90 | 51.78 |
| GRAPE-M-nonctx | 56.82 | 28.84 | 44.36 | 32.80 | 68.93 | **53.67** | 77.10 | 51.79 |
| AliBi | 58.21 | 29.78 | 45.38 | 34.60 | **70.08** | 53.51 | 78.50 | 52.87 |
| FoX | 58.38 | 30.89 | **45.80** | 35.00 | 69.37 | 52.88 | 78.40 | 52.96 |
| GRAPE-A-Q | 58.00 | 30.89 | 45.70 | **36.20** | 68.93 | 52.96 | 77.00 | 52.81 |
| GRAPE-A-K | 58.54 | 30.72 | 45.13 | 34.60 | 68.39 | 53.35 | 76.30 | 52.43 |
| GRAPE-A-QK | 57.95 | **32.00** | 45.77 | 33.40 | 69.37 | 53.51 | 79.00 | 53.00 |
| GRAPE-AP | **59.26** | 31.31 | 45.42 | 35.60 | 68.17 | 53.28 | **79.70** | **53.25** |
| RoPE (w/ KV-shift) | 58.04 | 30.80 | 45.04 | 35.40 | 68.39 | 53.99 | 78.60 | 52.89 |
| AliBi (w/ KV-shift) | **59.51** | **31.31** | 45.93 | 35.40 | 69.15 | 52.88 | 78.10 | 53.18 |
| FoX (w/ KV-shift) | 58.84 | 30.97 | **46.25** | 34.20 | 67.85 | 55.41 | **79.70** | 53.32 |
| GRAPE-AP (w/ KV-shift) | 57.32 | 30.55 | 46.18 | **35.80** | 69.10 | **55.64** | 79.60 | **53.46** |

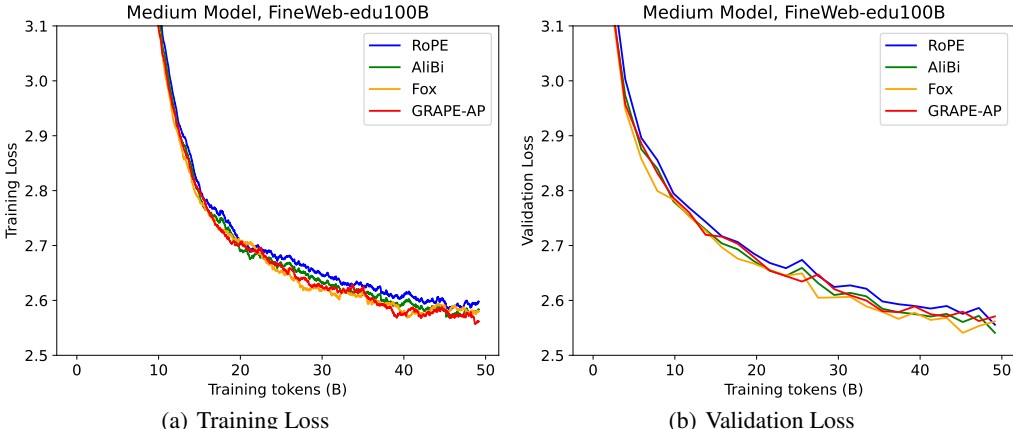

(a) Training Loss        (b) Validation Loss

Figure 2: The training and validation loss of medium-size models (353M), with different positional encoding mechanisms on the FineWeb-Edu 100B dataset.

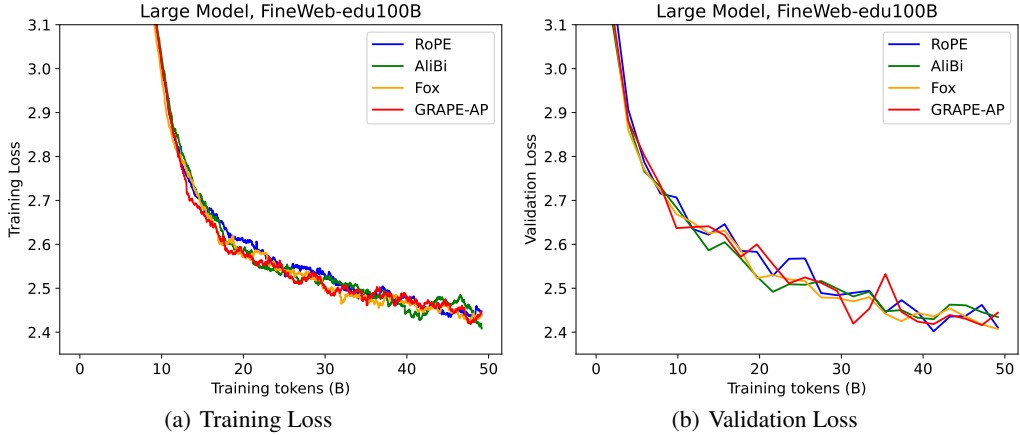

(a) Training Loss

(b) Validation Loss

Figure 3: The training and validation loss of large-size models (770M), with different positional encoding mechanisms on the FineWeb-Edu 100B dataset.

Table 2: The evaluation results of large models with different positional encoding mechanisms pretrained using the FineWeb-Edu 100B dataset (0-shot with lm-evaluation-harness). The best scores in each column are **bolded**, and the second best are underlined. Results are ranked separately for w/o KV-shift and w/ KV-shift.

| Method | ARC-E | ARC-C | HellaSwag | OBQA | PIQA | WinoGrande | SciQ | Avg. |
|---|---|---|---|---|---|---|---|---|
| RoPE | 62.63 | 32.76 | 51.01 | 36.40 | 71.33 | 55.72 | 80.50 | 55.76 |
| GRAPE-M-ctx | 60.52 | 32.51 | 50.13 | 35.40 | 70.67 | 54.38 | 79.50 | 54.73 |
| GRAPE-M-nonctx | 60.61 | 33.45 | 49.47 | 35.60 | 70.51 | 53.35 | 80.70 | 54.81 |
| AliBi | 62.67 | **34.39** | 51.33 | 36.60 | 71.11 | 56.27 | 82.70 | 56.44 |
| FoX | 61.07 | 33.11 | **51.85** | **37.80** | 71.27 | 55.33 | 83.70 | 56.30 |
| GRAPE-AP | **63.89** | 34.22 | 51.52 | 35.40 | **71.98** | **56.99** | **84.40** | **56.91** |
| RoPE (w/ KV-shift) | 62.12 | 33.70 | 50.13 | 35.80 | 71.49 | 53.51 | 80.50 | 55.32 |
| AliBi (w/ KV-shift) | 63.68 | **34.56** | 52.04 | 37.80 | 71.38 | 55.09 | **83.90** | 56.92 |
| FoX (w/ KV-shift) | 63.55 | 33.96 | **52.72** | 38.40 | **71.71** | **56.12** | 83.20 | **57.09** |
| GRAPE-AP (w/ KV-shift) | **63.72** | 33.11 | 52.29 | **39.00** | 71.65 | 54.78 | 83.50 | 56.86 |

## 7 CONCLUSION

GRAPE provides a general framework for positional encoding based on group actions, unifying *multiplicative* and *additive* mechanisms. Multiplicative GRAPE offers a closed-form, rank-2 exponential that is relative, compositional, and norm-preserving; it recovers RoPE and yields learned-basis and non-commuting extensions at controlled cost. Additive GRAPE realizes ALiBi and FoX exactly via unipotent general linear group GL lifts with the same streaming/cache policy. The GRAPE framework integrates seamlessly with existing Transformer models and offers a principled, extensible design space for future architectures.

## ACKNOWLEDGEMENTS

We thank the anonymous reviewers and area chairs for their helpful comments. We also thank Zhixuan Lin, Songlin Yang, and others for their constructive feedback on the experimental settings. We used Large Language Models (LLMs) as assistive tools to polish part of this paper, and the roles of LLMs in this work are restricted to improving presentation.

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

# Appendix

Table 3: Summary of Notation and Definitions.

| Symbol | Definition |
|---|---|
| $\mathrm{GL}(d)$ | **General Linear Group**: The group of all $d \times d$ invertible matrices. |
| $\mathrm{SO}(d)$ | **Special Orthogonal Group**: The group of $d \times d$ orthogonal matrices with determinant 1 ($\mathbf{R}^\top \mathbf{R} = \mathbf{I}$, $\det(\mathbf{R}) = 1$). |
| $\mathrm{SL}(d)$ | **Special Linear Group**: The group of $d \times d$ matrices with determinant 1. |
| $\mathfrak{gl}(d)$ | **general linear algebra**: The Lie algebra of $\mathrm{GL}(d)$, consisting of all $d \times d$ matrices. |
| $\mathfrak{so}(d)$ | **special orthogonal algebra**: The Lie algebra of $\mathrm{SO}(d)$, consisting of all $d \times d$ skew-symmetric matrices ($\mathbf{L}^\top = -\mathbf{L}$). |
| $\exp(\cdot)$ | **Exponential Map**: A map from a Lie algebra (generator) to a Lie group (operator). |
| $\mathbf{R}_2(\theta)$ | **2D Rotation Matrix**: The matrix $\begin{pmatrix} \cos\theta & -\sin\theta \\ \sin\theta & \cos\theta \end{pmatrix}$. |
| $\mathbf{G}(n)^\top$ | **Transpose (in $\mathrm{SO}(d)$)**: For $\mathbf{G} \in \mathrm{SO}(d)$, the transpose is the group inverse ($\mathbf{G}^\top = \mathbf{G}^{-1}$). |
| $\mathbf{G}(n)^{-\top}$ | **Inverse Transpose (in $\mathrm{GL}(d)$)**: The transpose of the matrix inverse, $(\mathbf{G}^{-1})^\top$. |
| Unipotent | **Unipotent Transform**: A linear transformation whose eigenvalues are all 1. |
| $\mathbf{p}_{u,h}$ | **Positional Embedding/Representation**: A vector derived from token-local features, obtained via a linear projection followed by RMS normalization. |

## A    RELATED WORK

Positional information in Transformers mainly can be categorized into these classes: (a) absolute encodings (sinusoidal or learned) (Vaswani et al., 2017; Devlin et al., 2019; Neishi and Yoshinaga, 2019; Kiyono et al., 2021; Likhomanenko et al., 2021; Wang et al., 2020; Liu et al., 2020; Wang et al., 2021; Sinha et al., 2022; Wennberg and Henter, 2021; Ke et al., 2020); (b) relative encodings that depend on offsets (Shaw et al., 2018; Dai et al., 2019; Raffel et al., 2020; He et al., 2020); and (c) linear logit biases with strong length extrapolation (Press et al., 2021; Chi et al., 2022a;b; Li et al., 2023; Ruoss et al., 2023), all shaping recency/extrapolation behavior (Haviv et al., 2022; Kazemnejad et al., 2023).

**Multiplicative position encoding.** RoPE realizes offsets as block-diagonal planar rotations of queries/keys, preserving norms and exact origin invariance; it is widely deployed across LLMs and modalities (Su et al., 2024; Touvron et al., 2023a;b; Heo et al., 2024). Angle/spectrum designs improve long-context fidelity (e.g., xPos) (Sun et al., 2022); LRPE formalizes separable relative transforms for linear attention models (Qin et al., 2023); mechanistic work analyzes frequency usage (Barbero et al., 2024). These methods are also compatible with sparse/linear attentions (Beltagy et al., 2020; Zaheer et al., 2020; Katharopoulos et al., 2020; Choromanski et al., 2020) and with context-scaling procedures (Xiong et al., 2023; Chen et al., 2023; Peng et al., 2023; Zhu et al., 2023; Jin et al., 2024). Beyond 1D language modeling, 2D RoPE and variants adapt rotary encodings to 2D grids by applying rotations along spatial axes, and have been shown to improve high-resolution extrapolation in Vision Transformers and related vision models (Heo et al., 2024). Recently, LieRE (Ostmeier et al., 2025) learns dense skew-symmetric generators whose exponentials produce high-dimensional rotations for multi-modal, $n$-dimensional inputs, while STRING (Schenck et al., 2025) designs separable, translation-invariant RoPE-style encodings that scale to 2D and 3D coordinates in vision and robotics settings (Ostmeier et al., 2025; Schenck et al., 2025). **GRAPE-M** identifies RoPE as commuting rank-2 exponentials in $\mathrm{SO}(d)$ and extends it to learned subspaces and compact non-commuting mixtures in closed form and a much faster way. Compared with LieRE, which parameterizes a dense skew-symmetric generator and applies a numerical matrix exponential (e.g., `torch.matrix_exp`) with $\mathcal{O}(d^3)$ time and $\mathcal{O}(d^2)$ parameters per head, Multiplicative GRAPE decomposes the action into rank-2 subspaces and uses the closed-form Rodrigues-type formulas from Section 2.3, so we only need vector–vector operations with $\mathcal{O}(d)$ cost per head (a detailed comparison between LieRE and GRAPE is presented in Appendix F.)

**Additive position encoding and forgetting mechanisms.** Additive schemes such as ALiBi (Press et al., 2021) and related kernelized/randomized forms (Chi et al., 2022a;b; Li et al., 2023; Ruoss et al., 2023) are captured exactly by GRAPE-A as unipotent actions in the general linear group GL that preserve the same relative law and streaming cacheability. Importantly, *forgetting mechanisms are additive*: the Forgetting Transformer (FoX) implements a learnable per-head exponential decay in the attention logits and is a specific GRAPE-A / GRAPE-AP instance imposing distance-dependent attenuation (Lin et al., 2025). FoX's data-dependent forget gates yield a path-additive bias $\mathbf{D}$ that we show is exactly the endpoint-independent GRAPE-AP case; see Appendix C for a constructive equivalence and its streaming implementation (Lin et al., 2025).

**Contextual position encoding.** Content-adaptive position modulates effective phase or distance via token features through gating/scaling and algebraic parameterizations (Wu et al., 2020; Zheng et al., 2024; Kogkalidis et al., 2024), and contextual counting (CoPE) (Golovneva et al., 2024). GRAPE introduces phase-modulated and dictionary-based contextual variants that replace a linear phase with cumulative token-adaptive phases (single or multi-subspace) while retaining exact headwise relativity and streaming caches. Finally, models can length-generalize without explicit encodings ("NoPE") under suitable training (Wang et al., 2024), which corresponds to the trivial generator $\mathbf{L} = 0$ in our view.

# B    APPLICATION IN MULTI-HEAD ATTENTION

Building upon the algebraic foundation for relative encoding established in Section 2.5, this section details the concrete integration of the rotational map $\mathbf{G}(n)$ into the full Multi-Head Attention (MHA) architecture, covering the per-head formulation, streaming policy, and implementation complexity.

**Per-head formulation.** Let $H$ be the number of heads and $d$ the per-head width. For head $h \in [H]$, let $(\mathbf{q}_{t,h}, \mathbf{k}_{t,h}, \mathbf{v}_{t,h}) \in \mathbb{R}^d$ denote the query/key/value at position $t$. A **GRAPE-M** position map is realized as an orthogonal operator $\mathbf{G}_{h,t} \in \mathrm{SO}(d)$ applied to $(\mathbf{q}_{t,h}, \mathbf{k}_{t,h})$:

$$\widetilde{\mathbf{q}}_{t,h} = \mathbf{G}_{h,t}\, \mathbf{q}_{t,h}, \qquad \widetilde{\mathbf{k}}_{t,h} = \mathbf{G}_{h,t}\, \mathbf{k}_{t,h}, \qquad \widetilde{\mathbf{v}}_{t,h} = \mathbf{v}_{t,h}.$$

The headwise attention logits and outputs are then

$$\ell_{t,j,h} = \frac{\widetilde{\mathbf{q}}_{t,h}^{\top}\widetilde{\mathbf{k}}_{j,h}}{\sqrt{d}} = \frac{\mathbf{q}_{t,h}^{\top}\big(\mathbf{G}_{h,t}^{\top}\mathbf{G}_{h,j}\big)\mathbf{k}_{j,h}}{\sqrt{d}}, \qquad \mathbf{y}_{t,h} = \sum_{j \leq t} \mathrm{softmax}\big(\ell_{t,\cdot,h}\big)_j\, \widetilde{\mathbf{v}}_{j,h}, \qquad \text{(B.1)}$$

with the usual output projection applied after concatenation across heads.

**Exact relative law.** If $\mathbf{G}_{h,t}$ arises from a one-parameter subgroup $\mathbf{G}_h(n) = \exp(n\, \mathbf{L}_h)$ (commuting MS-GRAPE-M, including RoPE and learned commuting bases), then

$$\mathbf{G}_{h,t}^{\top}\mathbf{G}_{h,j} = \mathbf{G}_h(j-t) \qquad \implies \qquad \ell_{t,j,h} = \frac{\mathbf{q}_{t,h}^{\top}\mathbf{G}_h(j-t)\, \mathbf{k}_{j,h}}{\sqrt{d}},$$

so logits depend only on the offset $j-t$ (exact origin invariance).

**Streaming cache.** Applying the rotational map $\mathbf{G}(t)$ independently to each query and key vector is the core property that enables an efficient streaming cache policy. For any type where $\mathbf{G}_t$ is known at token arrival (non-contextual and phase-modulated), cache $\widetilde{\mathbf{k}}_{j,h} = \mathbf{G}_{h,j}\mathbf{k}_{j,h}$ once and never rewrite it; at step $t$, compute $\widetilde{\mathbf{q}}_{t,h} = \mathbf{G}_{h,t}q_{t,h}$ and use logits $\ell_{t,j,h} = \widetilde{\mathbf{q}}_{t,h}^{\top}\widetilde{\mathbf{k}}_{j,h}/\sqrt{d}$.

# C    FORGETTING TRANSFORMER AS A SPECIAL ADDITIVE GRAPE

The Forgetting Transformer (FoX) introduces a scalar forget gate $f_t \in (0, 1]$ per head and timestep and adds the cumulative log-gate as an additive bias in the attention logits. Concretely, for a head $h$,

$$f_{t,h} = \sigma(\mathbf{w}_{f,h}^{\top}\mathbf{x}_t + b_{f,h}), \qquad F_{ij,h} = \prod_{\ell=j+1}^{i} f_{\ell,h}, \qquad D_{ij,h} = \log F_{ij,h} = \sum_{\ell=j+1}^{i} \log f_{\ell,h},$$

and the attention is

$$\mathbf{O}_h = \mathrm{softmax}\Big(\tfrac{1}{\sqrt{d}}\mathbf{Q}\mathbf{K}^{\top} + \mathbf{D}_h\Big)\mathbf{V}. \qquad \text{(FoX)}$$

We now show that Eq. (FoX) is exactly realized by our GRAPE-A framework using the endpoint-independent path-additive specialization of Section 5.

**FoX as GRAPE-AP with endpoint-independent edges.** In GRAPE-AP (Section 5), a head-wise additive logit $b_h(t, j)$ arises as a causal path sum

$$b_h(t, j) = \sum_{\ell=j+1}^{t} \psi_h(t, \ell).$$

If the edge potentials do not depend on the endpoint, i.e. $\psi_h(t, \ell) \equiv a_{\ell,h}$, then $b_h(t, j)$ reduces to a difference of per-time potentials:

$$b_h(t, j) = \sum_{\ell=j+1}^{t} a_{\ell,h} = U_{t,h} - U_{j,h}, \qquad U_{u,h} := \sum_{\ell<u} a_{\ell,h}.$$

FoX corresponds to the choice $a_{\ell,h} = \log f_{\ell,h} \leq 0$, yielding

$$b_h(t, j) \equiv D_{ij,h} = \sum_{\ell=j+1}^{t} \log f_{\ell,h}.$$

Thus, the FoX forgetting bias $\mathbf{D}_h$ is precisely the GRAPE-AP path-integral additive bias with endpoint-independent edges.

**Unipotent GL lift (GRAPE-A view).** Let $\mathbf{E} := \mathbf{e}_{d+2}\mathbf{e}_{d+1}^\top$ be the rank-1 nilpotent used in Section 4.2. For a fixed head $h$ and endpoint $t$, define per-link unipotent factors

$$\mathbf{H}_h^{(t)}(\ell) = \mathbf{I} + \psi_h(t, \ell)\,\mathbf{E}, \qquad \psi_h(t, \ell) = \log f_{\ell,h}.$$

Since $\mathbf{E}^2 = \mathbf{0}$, the path product collapses:

$$\prod_{\ell=j+1}^{t} \mathbf{H}_h^{(t)}(\ell) = \mathbf{I} + \Big( \sum_{\ell=j+1}^{t} \log f_{\ell,h} \Big) \mathbf{E} = \mathbf{I} + D_{ij,h}\,\mathbf{E}.$$

Scoring in homogeneous coordinates as in Section 4 with the paired inverse-transpose,

$$\widetilde{\mathbf{q}}_{t,h}^\top \widetilde{\mathbf{k}}_{j,h} = \widehat{\mathbf{q}}_{t,h}^\top \Big( \mathbf{I} + D_{ij,h}\,E \Big)^{-\top} \widehat{\mathbf{k}}_{j,h} = \mathbf{q}_{t,h}^\top \mathbf{k}_{j,h} + D_{ij,h},$$

recovers Eq. (FoX) exactly (up to the standard $1/\sqrt{d}$ factor we include throughout). Hence, FoX is an exact GRAPE-A / GRAPE-AP instance realized by a rank-1 unipotent path with endpoint-independent edges.

**Streaming and complexity.** Compute prefix sums $U_{t,h} = \sum_{\ell<t} \log f_{\ell,h}$ once per step; then $D_{ij,h} = U_{i,h} - U_{j,h}$ is obtained by subtraction, preserving the $O(L)$ rowwise cost and the streaming cache policy from Section 4–Section 5. The headwise gates $f_{t,h}$ add $O(1)$ parameters and negligible computation.

**Special cases and composition.** If $f_{t,h} \equiv e^{-\beta_h}$ (constant per head), then $D_{ij,h} = -\beta_h(i-j)$ and FoX reduces to exact ALiBi (Section 4.2). More generally, FoX composes additively with the multiplicative (orthogonal) GRAPE acting on $(\mathbf{q}, \mathbf{k})$ as in Eq. (5.3), preserving norm-preservation of the rotational part while adding bounded, non-positive, content-adaptive path biases.

## D  NON-COMMUTING MULTIPLICATIVE GRAPE

Consider the thin compression $\mathbf{L} = \mathbf{E}\mathbf{L}_r\mathbf{E}^\top$ with $\mathbf{E} \in \mathbb{R}^{d \times r}$ orthonormal and $\mathbf{L}_r \in \mathfrak{so}(r)$. Then

$$\sigma(\mathbf{L}) = \sigma(\mathbf{L}_r) \cup \{0\}^{d-r}, \qquad \sigma\big( \exp(n\mathbf{L}) \big) = \sigma\big( \exp(n\mathbf{L}_r) \big) \cup \{1\}^{d-r}.$$

If $\mathbf{L}_r = \mathbf{T}(\bigoplus_{t=1}^{r/2} \theta_t \mathbf{J})\mathbf{T}^\top$ is the real-Schur form, then the nontrivial eigenvalues are $\{\pm i\theta_t\}_{t=1}^{r/2}$ and $e^{\pm in\theta_t}$ for the exponential. Thus, the expressive power of non-contextual non-commuting MS-GRAPE is fully captured by the $r/2$ mode angles $\{\theta_t\}$; the ambient lifting via $\mathbf{E}$ preserves the spectrum.

## E    COMPOSITION OF ADDITIVE GRAPE AND MULTIPLICATIVE GRAPE

For the unipotent forms of Additive GRAPE, applying the inverse transpose $\mathbf{G}_{\mathrm{add}}(m)^{-\top}$ requires only simple scalar-vector operations per active component. Thus, the per-head overhead is $O(d)$ and is negligible relative to attention matrix multiplications.

Multiplicative GRAPE (Section 3) and Additive GRAPE (Section 4) compose naturally. This can be viewed either additively at the logit level:

$$\ell_{t,j,h} = \tfrac{1}{\sqrt{d}}\,\mathbf{q}_{t,h}^{\top}\mathbf{G}_h(j-t)\mathbf{k}_{j,h} \;+\; \underbrace{(j-t)\,\omega\,\big(\lambda_{q,t}+\lambda_{k,j}\big)}_{\text{Additive GRAPE bias}},$$

or rigorously as a single group action in the general linear group $\mathrm{GL}(d{+}2)$. Concretely, we construct the joint lift by direct sum. Let $\widehat{\mathbf{q}} = [\mathbf{q};1;0]$ and $\widehat{\mathbf{k}} = [\mathbf{k};0;1]$ be the augmented vectors. Define the joint generator $\mathbb{L} \in \mathfrak{gl}(d{+}2)$ as the direct sum of the rotational generator $\mathbf{L} \in \mathfrak{so}(d)$ and the gated nilpotent generator $\omega\Lambda\mathbf{A}_0$ (where $\Lambda = \lambda_q + \lambda_k$ captures the content modulation):

$$\mathbb{G}_{\mathrm{joint}}(m) \;=\; \exp(m\mathbb{L}) \;=\; \begin{bmatrix} \exp(m\mathbf{L}) & \mathbf{0} & \mathbf{0} \\ \mathbf{0}^{\top} & 1 & m\,\omega\,\Lambda \\ \mathbf{0}^{\top} & 0 & 1 \end{bmatrix} \in \mathrm{GL}(d{+}2).$$

This block-diagonal structure unifies the mechanisms: the top-left block applies the norm-preserving RoPE rotation to the features, while the bottom-right unipotent block generates the additive scalar bias via the auxiliary coordinates. Scoring with the paired inverse-transpose strictly preserves the exact relative law for the combined system:

$$\widehat{\mathbf{q}}_i^{\top}\,\mathbb{G}_{\mathrm{joint}}(j{-}i)^{-\top}\,\widehat{\mathbf{k}}_j \;=\; \mathbf{q}_i^{\top}\exp\big((j{-}i)\mathbf{L}\big)\mathbf{k}_j \;+\; (j{-}i)\,\omega\,\Lambda \;+\; \mathrm{const},$$

exactly reproducing the sum of multiplicative (rotary) and additive (slope) components. In this unified view, both mechanisms are simply projections of a single one-parameter subgroup action in high-dimensional space, retaining exact relativity and efficient streaming caches.

## F    COMPARISON WITH LIERE

Lie Rotational Position Encodings (LieRE) (Ostmeier et al., 2025) encode positional information by learning a skew-symmetric generator in $\mathrm{SO}(d)$. The method then applies the matrix exponential of this generator to get a rotational position map. For each attention head, the method learns one skew matrix. Its exponential gives a dense orthogonal operator on queries and keys. Positions then match elements of a one-parameter subgroup on the rotation manifold. This picture is a compact Lie theoretic version of RoPE style encodings. Different heads can learn distinct rotational geometries, and the map keeps the norm and an exact relative position law.

Formally, for head $h$ the generator is $G_h \in \mathfrak{so}(d)$. The positional map is $x \mapsto \exp(n\omega_h G_h)x$. A direct implementation has cost $T_{\mathrm{LieRE}}(d) = \Theta(d^3)$ per head for the matrix exponential and needs $\Theta(d^2)$ parameters and the same order of memory.

Multiplicative GRAPE and LieRE both use rotations in $\mathrm{SO}(d)$ that come from skew-symmetric generators. LieRE gives each head a dense or block skew matrix. It forms the positional operator with the full matrix exponential $\exp(G)$. This creates very rich rotations but needs $\mathcal{O}(d^3)$ time for the exponential and $\mathcal{O}(d^2)$ parameters and memory per head. GRAPE-M restricts the generator to a sum of rank 2 planes and uses a closed form Rodrigues-type formula for the exponential (Section 2). For one token, the positional mapping then reduces to a few inner products and vector updates. So the cost is $\mathcal{O}(d)$ time and $\mathcal{O}(d)$ memory per head.

This choice of parametrization has two main effects in practice. First, the GRAPE-M scale cleanly translates to contextual versions where frequencies or phases depend on the token content. The closed-form expression can be computed quickly for each token, and there is no large matrix exponential. In the LieRE setup, one needs a new dense matrix exponential for each content-dependent generator. This step is much more costly and makes such contextual use harder to deploy in real models. Second, GRAPE gives a single group-theoretic picture for multiplicative and additive mechanisms. The multiplicative part lives in $\mathrm{SO}(d)$ and additive or forgetting style terms (ALiBi, FoX, GRAPE-A, GRAPE-AP) come from unipotent actions in $\mathrm{GL}$ with the same relative law and the same streaming cacheability (Sections 4-5). LieRE only targets rotational encodings and does not model additive logit biases or forgetting terms.

# G  2D AND 3D GRAPE FOR VISION AND MULTIMODAL POSITION ENCODING

Extending GRAPE beyond one-dimensional token positions is easy. The construction only needs a chosen group action on coordinates.

For images with integer pixel coordinates $(u, v) \in \mathbb{Z}^2$ we pick two generators $\mathbf{L}^{(x)}$ and $\mathbf{L}^{(y)}$. A token at $(u, v)$ then gets the encoding

$$\mathbf{G}_{2\mathrm{D}}(u, v) = \exp\big(u\,\omega_x\mathbf{L}^{(x)}\big)\,\exp\big(v\,\omega_y\mathbf{L}^{(y)}\big) \in \mathrm{SO}(d).$$

The two generators act on 2D planes that can be disjoint in the base design. In that case, the map reduces to a RoPE-style separable encoding. A learned choice of planes inside $\mathbb{R}^d$ gives the GRAPE-M variant again.

For 3D coordinates $(u, v, w)$ that mark video space time tokens or point clouds, we follow the same pattern. We introduce three commuting generators and define

$$\mathbf{G}_{3\mathrm{D}}(u, v, w) = \exp\big(u\,\omega_x\mathbf{L}^{(x)}\big)\,\exp\big(v\,\omega_y\mathbf{L}^{(y)}\big)\,\exp\big(w\,\omega_z\mathbf{L}^{(z)}\big).$$

In the non-commuting case, we use the thin Schur mode compression from Appendix D. The closed-form rank 2 matrix exponential from the main text still applies. The per token cost stays $\mathcal{O}(d)$ even for higher-dimensional coordinate spaces.

On the additive side, GRAPE-A and GRAPE-AP handle 2D or 3D structures through the scalar offset $m$. The value $m$ can be any function of coordinate differences. For an image, we can take

$$m = \alpha_x(u_t - u_j) + \alpha_y(v_t - v_j),$$

and this keeps the same algebraic template. For 3D settings, we can set

$$m = \|\mathbf{r}_t - \mathbf{r}_j\|$$

with $\mathbf{r}_t$ and $\mathbf{r}_j$ in $\mathbb{R}^3$. The update matrix then stays unipotent, and the exact relative composition law still holds. This gives a clear way to impose axis-aligned or radial recency bias in vision and multimodal models.

# H  ALGORITHMIC DETAILS AND PSEUDO CODE

This appendix contains the detailed pseudocode.

---

**Algorithm 1** Commuting Multi-Subspace GRAPE-M

---

**Require:** $\mathbf{Q}, \mathbf{K} \in \mathbb{R}^{B \times L \times H \times d}$, orthogonal $\mathbf{E} \in \mathbb{R}^{d \times d}$, frequencies $\{\omega_{h,j}\}_{j=1}^{d/2}$, positions $n \in \mathbb{Z}^L$
1: **for** $h = 1$ **to** $H$ **do**
2:     $\mathbf{Q}'[:, :, h, :] \leftarrow \mathbf{Q}[:, :, h, :]\,\mathbf{E};\quad \mathbf{K}'[:, :, h, :] \leftarrow \mathbf{K}[:, :, h, :]\,\mathbf{E}$
3:     **for** $\ell = 0$ **to** $L - 1$ **do**
4:         **for** $j = 1$ **to** $d/2$ **do**
5:             $\theta \leftarrow n_\ell\,\omega_{h,j}$; apply $2 \times 2$ rotation $\mathbf{G}_2(\theta)$ to coords $(2j-1, 2j)$ of $\mathbf{Q}'[:, \ell, h, :]$ and $\mathbf{K}'[:, \ell, h, :]$
6:         **end for**
7:     **end for**
8:     $\widetilde{\mathbf{Q}}[:, :, h, :] \leftarrow \mathbf{Q}'\,\mathbf{E}^\top;\quad \widetilde{\mathbf{K}}[:, :, h, :] \leftarrow \mathbf{K}'\,\mathbf{E}^\top$
9: **end for**
10: **return** $(\widetilde{\mathbf{Q}}, \widetilde{\mathbf{K}})$

---

# I  DIFFERENTIATION AND FAST APPLICATION OF RANK-2 MATRIX EXPONENTIAL

**Differentiation and stability.** Let $f_1(z) = \frac{\sin z}{z}$ and $f_2(z) = \frac{1 - \cos z}{z^2}$ with $z = n\omega s$. Then

$$\exp(n\omega\mathbf{L}) = \mathbf{I} + f_1(z)\mathbf{L} + f_2(z)\mathbf{L}^2.$$

For any scalar parameter $\theta \in \{\omega\} \cup \{\text{entries of } a, b\}$,

$$\partial_\theta \exp(n\omega \mathbf{L}) = f_1(z)\,\partial_\theta \mathbf{L} + f_2(z)\,(\mathbf{L}\,\partial_\theta \mathbf{L} + \partial_\theta \mathbf{L}\,\mathbf{L}) + \partial_\theta z\,\left(f_1'(z)\mathbf{L} + f_2'(z)\mathbf{L}^2\right),$$

$$\partial_\theta z = n\omega\,\partial_\theta s + ns\,\partial_\theta \omega, \qquad \partial_\theta s = \tfrac{1}{2}s^{-1}\partial_\theta(\alpha\beta - \gamma^2).$$

Use series for $|z| < \varepsilon$: $f_1(z) = 1 - \frac{z^2}{6} + O(z^4)$ and $f_2(z) = \frac{1}{2} - \frac{z^2}{24} + O(z^4)$. These formulas enable mixed-precision backprop with small-$s$ guards.

**Fast application.** For any $x \in \mathbb{R}^d$,

$$\mathbf{L}\mathbf{x} = \mathbf{a}\langle \mathbf{b}, \mathbf{x}\rangle - \mathbf{b}\langle \mathbf{a}, \mathbf{x}\rangle, \qquad \mathbf{L}^2\mathbf{x} = \gamma(\mathbf{a}\langle \mathbf{b}, \mathbf{x}\rangle + \mathbf{b}\langle \mathbf{a}, \mathbf{x}\rangle) - \beta\,\mathbf{a}\langle \mathbf{a}, \mathbf{x}\rangle - \alpha\,\mathbf{b}\langle \mathbf{b}, \mathbf{x}\rangle.$$

Thus $\mathbf{G}(n)\mathbf{x} = \mathbf{x} + f_1\mathbf{L}\mathbf{x} + f_2\mathbf{L}^2\mathbf{x}$ with $f_1 = \frac{\sin(n\omega s)}{s}$ and $f_2 = \frac{1-\cos(n\omega s)}{s^2}$, which is evaluable in $O(d)$ time via a few inner products. By the minimal polynomial $\lambda(\lambda^2 + s^2)$, $\mathbf{L}^3 = -s^2\mathbf{L}$; expanding $\exp(\eta\mathbf{L})$ and regrouping yields the rank-2 update form used throughout

# J  SPECTRAL ANALYSIS OF GRAPE AND OTHER METHODS

In this section, we discuss eigenvalue-level results for GRAPE-M generators/exponentials and summarize the unipotent spectra of GRAPE-A/GRAPE-AP. Throughout, $\mathbf{L}(\mathbf{a}, \mathbf{b}) = \mathbf{a}\mathbf{b}^\top - \mathbf{b}\mathbf{a}^\top \in \mathfrak{so}(d)$, and $\alpha = \|\mathbf{a}\|^2$, $\beta = \|\mathbf{b}\|^2$, $\gamma = \mathbf{a}^\top\mathbf{b}$, $\Delta = \alpha\beta - \gamma^2$, $s = \sqrt{\Delta}$ as in Section 2.

## J.1  RANK-2 PLANE: EXACT SPECTRUM AND GEOMETRIC INTERPRETATION

**Lemma J.1** (Rank-2 spectrum). For $\mathbf{L} = \mathbf{L}(\mathbf{a}, \mathbf{b})$, the eigenvalues are $\{\pm is\} \cup \{0\}^{d-2}$, and there exists $\mathbf{B} \in \mathrm{SO}(d)$ such that

$$\mathbf{B}^\top\mathbf{L}\mathbf{B} = \begin{bmatrix} s\mathbf{J} & \mathbf{0} \\ \mathbf{0} & \mathbf{0}_{d-2} \end{bmatrix}, \qquad \mathbf{J} = \left(\begin{smallmatrix} 0 & -1 \\ 1 & 0 \end{smallmatrix}\right).$$

Moreover, $s = \|\mathbf{a}\|\|\mathbf{b}\|\sin\phi$, where $\phi \in [0, \pi]$ is the angle between $a$ and $b$.

*Proof.* From Section 2, $\mathbf{L}^2 = -s^2\mathbf{P}_\mathcal{U}$ with $\mathcal{U} = \mathrm{span}\{\mathbf{a}, \mathbf{b}\}$, whence the minimal polynomial is $\lambda(\lambda^2 + s^2)$ and $\sigma(\mathbf{U}) = \{\pm is, 0\}$. Choosing an orthonormal basis aligned with $\mathcal{U} \oplus \mathcal{U}^\perp$ yields the claimed form. Finally, $\Delta = \alpha\beta - \gamma^2 = \|\mathbf{a}\|^2\|\mathbf{b}\|^2(1 - \cos^2\phi) = (\|\mathbf{a}\|\|\mathbf{b}\|\sin\phi)^2$. $\qquad\square$

**Corollary J.2** (Phase bounds and orthogonality). The per-step rotation angle of $\exp(\eta\mathbf{L})$ on $\mathcal{U}$ equals $\theta = \eta s$ and satisfies $0 \le \theta \le \eta\|\mathbf{a}\|\|\mathbf{b}\|$, with equality when $\mathbf{a} \perp \mathbf{b}$. If $\mathbf{b} = \mathcal{J}\mathbf{a}$ (Section 2.4) and $\|\mathbf{a}\| = 1$, then $s = 1$ and $\theta = \eta$.

**Exponential spectrum.** For any $n \in \mathbb{Z}$,

$$\sigma\big(\exp(n\mathbf{L})\big) = \{e^{\pm ins}\} \cup \{1\}^{d-2}.$$

Hence $\rho(\exp(n\mathbf{L})) = 1$, the map is unitary (orthogonal), and all Lyapunov exponents are zero. Periodicity holds with fundamental period $T = 2\pi/s$ when $s/\pi \in \mathbb{Q}$; otherwise, the trajectory is quasi-periodic on the unit circle.

## J.2  MULTI-SUBSPACE GRAPE-M AND RoPE

Let $\mathbf{L} = \sum_{j=1}^m \theta_j\mathbf{L}_j$ with mutually orthogonal planes (hence $[\mathbf{L}_i, \mathbf{L}_j] = 0$ for $i \ne j$) and $\mathbf{L}_j = \mathbf{U}_j\mathbf{J}\mathbf{U}_j^\top$. Then

$$\mathbf{B}^\top\mathbf{L}\mathbf{B} = \bigoplus_{j=1}^m \theta_j\mathbf{J} \oplus \mathbf{0}_{d-2m}, \qquad \sigma(\mathbf{L}) = \{\pm i\theta_j\}_{j=1}^m \cup \{0\}^{d-2m},$$

for some $\mathbf{B} \in \mathrm{SO}(d)$. Consequently,

$$\sigma\big(\exp(n\mathbf{L})\big) = \{e^{\pm in\theta_j}\}_{j=1}^m \cup \{1\}^{d-2m}.$$

This recovers RoPE when the planes are the coordinate pairs and $\{\theta_j\}$ follow the canonical log-uniform spectrum (Proposition 3.1).

## J.3 ADDITIVE GRAPE

We now analyze the spectral properties of the additive lifts in GL introduced in Sections 4 and 5. The key structural fact is unipotency: all per-step factors are identity plus a rank-1 (or few-rank) nilpotent update of index 2.

**Setup.** Let $\mathbf{A} \in \mathfrak{gl}(d+1)$ (or $\mathfrak{gl}(d+2)$ for ALiBi) satisfy $\mathbf{A}^2 = \mathbf{0}$ as in Eq. (4.1) and Eq. (4.7). For a scalar path parameter $s \in \mathbb{R}$, define the unipotent factor

$$\mathbf{H}(s) := \exp(s\mathbf{A}) = \mathbf{I} + s\,\mathbf{A}, \qquad \mathbf{H}(s)^{-1} = \mathbf{I} - s\,\mathbf{A}, \qquad \det \mathbf{H}(s) = 1.$$

For Additive GRAPE(GRAPE-A) with offset $m = j-i$, $s = m\,\omega$; for GRAPE-PA, $s = s_h(t,j) := \sum_{\ell=j+1}^{t} \psi_h(t,\ell)$ from Eq. (5.2).

**Proposition J.3** (Eigenvalues and Jordan structure of additive lifts). Let $\mathbf{A} \in \mathfrak{gl}(D)$ satisfy $\mathbf{A}^2 = \mathbf{0}$ and $\mathbf{A} \neq \mathbf{0}$. Then for every $s \neq 0$,

$$\sigma\big(\mathbf{H}(s)\big) = \{1\}^D, \qquad (\mathbf{H}(s) - \mathbf{I})^2 = \mathbf{0}, \qquad \det \mathbf{H}(s) = 1, \qquad \rho(\mathbf{H}(s)) = 1.$$

Hence, the minimal polynomial of $\mathbf{H}(s)$ is $(\lambda - 1)^2$, and the Jordan form consists of size-2 Jordan blocks for the 1-eigenspace, with the number of nontrivial blocks equal to $\mathrm{rank}(\mathbf{A})$.

*Proof.* Since $\mathbf{A}^2 = \mathbf{0}$, $\exp(s\mathbf{A}) = \mathbf{I} + s\mathbf{A}$ and $(\mathbf{H}(s) - \mathbf{I})^2 = s^2\mathbf{A}^2 = \mathbf{0}$. The characteristic polynomial is $(\lambda - 1)^D$ for $\mathbf{H}(s)$, so all eigenvalues equal 1. The determinant equals the product of eigenvalues, hence 1; the spectral radius is therefore 1. $\square$

**Dictionary closure.** If $\{\mathbf{A}_r\}_{r=1}^R$ satisfy $\mathbf{A}_r^2 = \mathbf{0}$ and $\mathbf{A}_r \mathbf{A}_s = \mathbf{0}$ for all $r, s$, then

$$\Big(\sum_r \theta_r \mathbf{A}_r\Big)^2 = \sum_r \theta_r^2 \mathbf{A}_r^2 + \sum_{r \neq s} \theta_r \theta_s \mathbf{A}_r \mathbf{A}_s = \mathbf{0},$$

so the combined generator is also index-2 nilpotent and yields the same unipotent spectrum.

**Singular values.** Although $\mathbf{H}(s)$ is not orthogonal, its deviation from $\mathbf{I}$ is rank-limited and exactly analyzable. We first give a sharp, explicit formula for the canonical rank-1 case (ALiBi block), then a general bound.

**Lemma J.4** (Exact singular-value pair for a canonical rank-1 unipotent). Let $\mathbf{E} := \mathbf{e}_p \mathbf{e}_q^\top$ with $p \neq q$ and define $\mathbf{H}(s) := \mathbf{I} + s\mathbf{E} \in \mathbb{R}^{D \times D}$. Then $D - 2$ singular values equal 1, and the remaining two are

$$\sigma_\pm(\mathbf{H}(s)) = \sqrt{1 + \tfrac{s^2}{2} \pm |s|\sqrt{1 + \tfrac{s^2}{4}}}, \qquad \sigma_+(\mathbf{H}(s))\,\sigma_-(\mathbf{H}(s)) = 1. \tag{J.1}$$

In particular,

$$\kappa_2(\mathbf{H}(s)) = \frac{\sigma_+(\mathbf{H}(s))}{\sigma_-(\mathbf{H}(s))} = \sigma_+(\mathbf{H}(s))^2 = 1 + \tfrac{s^2}{2} + |s|\sqrt{1 + \tfrac{s^2}{4}} = 1 + |s| + O(s^2) \quad \text{as } s \to 0.$$

*Proof.* The action of $\mathbf{H}(s)^\top \mathbf{H}(s)$ is identity on $\mathrm{span}\{\mathbf{e}_p, \mathbf{e}_q\}^\perp$. In the basis $\{\mathbf{e}_q, \mathbf{e}_p\}$ it equals $\begin{pmatrix} 1+s^2 & s \\ s & 1 \end{pmatrix}$, whose eigenvalues are $1 + \tfrac{s^2}{2} \pm |s|\sqrt{1 + \tfrac{s^2}{4}}$. Taking square roots yields (J.1). The product equals $\sqrt{\det(\mathbf{H}^\top \mathbf{H})} = |\det \mathbf{H}| = 1$. $\square$

**Corollary J.5** (ALiBi and Additive GRAPE(GRAPE-A) conditioning numbers). For the exact ALiBi generator in Eq. (4.7), let $\mathbf{E} := \mathbf{e}_{d+2} \mathbf{e}_{d+1}^\top$ so that $\mathbf{A}_h = -\beta_h \mathbf{E}$. Then $\mathbf{G}_{\mathrm{add},h}(m) = \mathbf{I} + m\mathbf{A}_h = \mathbf{I} - m\,\beta_h\,\mathbf{E} = \mathbf{I} + s\,\mathbf{E}$ with $s = -m\,\beta_h$, and the only nontrivial singular values follow from Eq. (J.1). For the single-vector additive lift Eq. (4.1) with $\mathbf{A} = \begin{pmatrix} \mathbf{0} & \mathbf{u}_{\mathrm{shift}} \\ \mathbf{0}^\top & 0 \end{pmatrix}$ and $\|\mathbf{u}_{\mathrm{shift}}\| = 1$, the same formula holds with $\mathbf{E}$ replaced by an orthogonally similar rank-1 update and $s = m\,\omega$.

**Lemma J.6** (General operator-norm bounds for index-2 unipotents). For any $\mathbf{A}$ with $\mathbf{A}^2 = \mathbf{0}$ and any $s \in \mathbb{R}$,

$$1 - |s| \, \|\mathbf{A}\|_2 \ \leq \ \sigma_{\min}(\mathbf{I} + s\mathbf{A}) \ \leq \ \sigma_{\max}(\mathbf{I} + s\mathbf{A}) \ \leq \ 1 + |s| \, \|\mathbf{A}\|_2.$$

These bounds are conservative but dimension-free. In the canonical rank-1 case of Lemma J.4 with $\|\mathbf{A}\|_2 = 1$, one has the sharper small-$|s|$ behavior $\sigma_{\max}(\mathbf{I} + s\mathbf{A}) = 1 + \frac{|s|}{2} + O(s^2)$ and $\sigma_{\min}(\mathbf{I} + s\mathbf{A}) = 1 - \frac{|s|}{2} + O(s^2)$.

*Proof.* Use the triangle inequality $\|(\mathbf{I} + s\mathbf{A})\mathbf{x}\|_2 \leq \|\mathbf{x}\|_2 + |s| \, \|\mathbf{A}\|_2 \|\mathbf{x}\|_2$ and its reverse form applied to $(\mathbf{I} + s\mathbf{A})^{-1} = \mathbf{I} - s\mathbf{A}$; see also Weyl inequalities for singular values under rank-1 perturbations. $\square$

**Cancellation in the relative logit.** While $\mathbf{H}(s)$ can be anisotropic (Lemma J.4), the Additive GRAPE(GRAPE-A) scoring uses a paired inverse-transpose (Eq. (4.2)), which cancels all multiplicative distortions and yields a pure additive term:

$$\widetilde{\mathbf{q}}_i^\top \widetilde{\mathbf{k}}_j = \widehat{\mathbf{q}}_i^\top \left(\mathbf{I} + i \, \omega \, \mathbf{A}\right)^\top \left(\mathbf{I} - j \, \omega \, \mathbf{A}^\top\right) \widehat{\mathbf{k}}_j \ = \ \widehat{\mathbf{q}}_i^\top \left(\mathbf{I} - (j{-}i) \, \omega \, \mathbf{A}^\top\right) \widehat{\mathbf{k}}_j \ = \ \widehat{\mathbf{q}}_i^\top \mathbf{G}_{\mathrm{add}}(j{-}i)^{-\top} \widehat{\mathbf{k}}_j,$$

since $(\mathbf{A}^\top)^2 = \mathbf{0}$. This reproduces the exact relative law Eq. (4.3) and the closed form Eq. (4.5) (e.g. Eq. (4.6)), independently of $\sigma_\pm(H(s))$.

**GRAPE-AP as a path-integral unipotent.** Fix a head $h$ and endpoint $t$. The per-row path product in Section 5 is

$$\prod_{\ell=j+1}^{t} \left(\mathbf{I} - \psi_h(t, \ell) \, \mathbf{E}\right) \ = \ \mathbf{I} - \left( \sum_{\ell=j+1}^{t} \psi_h(t, \ell) \right) \mathbf{E} \ = \ \mathbf{I} - s_h(t, j) \, \mathbf{E},$$

because $\mathbf{E}^2 = \mathbf{0}$. Thus GRAPE-AP inherits the unipotent spectrum of Prop. J.3 with row-dependent $s = s_h(t, j) \leq 0$ (since $\psi_h \leq 0$ by construction). Its only two nontrivial singular values are exactly (J.1) with $s \mapsto s_h(t, j)$; the rest equal 1. Consequently,

$$\kappa_2\big(\text{PA factor}\big) \ = \ \frac{\sigma_+\big({-}s_h(t,j)\big)}{\sigma_-\big({-}s_h(t,j)\big)} \ = \ \sigma_+\big({-}s_h(t,j)\big)^2$$

$$= \ 1 + \frac{s_h(t,j)^2}{2} + |s_h(t,j)| \sqrt{1 + \frac{s_h(t,j)^2}{4}} \ = \ 1 + |s_h(t,j)| + O\big(s_h(t,j)^2\big),$$

while the determinant remains 1 and eigenvalues are all 1. As in Additive GRAPE(GRAPE-A), the paired inverse-transpose used in the bilinear scoring removes any multiplicative anisotropy, leaving the bounded additive term $b_h(t, j)$ in Eq. (5.2).

**Implications.** Now we summarize the implications of previous results. For all $s$, $\mathbf{H}(s)$ is invertible with $\mathbf{H}(s)^{-1} = \mathbf{I} - s\mathbf{A}$; eigenvalues do not grow with offset length (spectral radius = 1). The operator norm grows at most linearly in $|s|$ (Lemma J.6) and is exactly characterized in the rank-1 canonical cases (Lemma J.4).

Secondly, $\det \mathbf{H}(s) = 1$ implies no net volume change; any expansion along one direction is exactly balanced by contraction along its paired direction (product $\sigma_+ \sigma_- = 1$). Despite anisotropy, the GRAPE-A and GRAPE-AP logits remain exactly relative because the key transform uses $\mathbf{H}(s)^{-\top}$, algebraically eliminating multiplicative distortion and yielding the closed-form additive bias (Eqs. (4.3), (4.5), (5.2)).

## J.4 Comparison to PaTH Attention

PaTH Attention (Yang et al., 2025b) proposes a contextual multiplicative position map given by a cumulative product of identity-plus-rank-one matrices

$$\mathbf{H}_t = \mathbf{I} - \beta_t \, \mathbf{w}_t \mathbf{w}_t^\top, \qquad \|\mathbf{w}_t\|_2 = 1, \quad \beta_t \in (0, 2),$$

applied along the path between key position $j$ and query position $i$ as $\prod_{s=j+1}^{i} \mathbf{H}_s$ (see Section 2 of the PaTH paper). In contrast to **GRAPE-M** factors, each $\mathbf{H}_t$ is *not* orthogonal unless $\beta_t \in \{0, 2\}$. This has immediate spectral consequences.

**Per-step spectrum.** Since $\mathbf{H}_t$ is symmetric rank-1 perturbation of the identity with projector $\mathbf{P}_t := \mathbf{w}_t \mathbf{w}_t^\top$,

$$\sigma(\mathbf{H}_t) = \{\, 1 - \beta_t, \underbrace{1, \ldots, 1}_{d-1} \,\}, \qquad \det(\mathbf{H}_t) = 1 - \beta_t, \qquad \|\mathbf{H}_t\|_2 = \max\{1, |1 - \beta_t|\} = 1.$$

Thus $\mathbf{H}_t$ is norm nonexpansive (operator norm 1) but *not norm-preserving* unless $\beta_t \in \{0, 2\}$. Singular values equal the absolute eigenvalues because $\mathbf{H}_t$ is symmetric; the component along $\mathbf{w}_t$ is scaled by $|1 - \beta_t| < 1$ for any $\beta_t \in (0, 2) \setminus \{0, 2\}$, and flips sign when $\beta_t > 1$ (a design choice in PaTH to allow negative eigenvalues for state-tracking).

**Path product is contractive and near-singular.** Let $\mathbf{P}_{j \to i} = \prod_{s=j+1}^{i} \mathbf{H}_s$. Submultiplicativity of singular values gives

$$\sigma_{\max}(\mathbf{P}_{j \to i}) \leq \prod_{s=j+1}^{i} \|\mathbf{H}_s\|_2 = 1, \qquad \sigma_{\min}(\mathbf{P}_{j \to i}) \geq \prod_{s=j+1}^{i} \sigma_{\min}(\mathbf{H}_s) = \prod_{s=j+1}^{i} |1 - \beta_s|.$$

Hence $\mathbf{P}_{j \to i}$ is (at best) nonexpansive, with a worst-case exponential lower bound on the smallest singular value governed by the path-length product of $|1 - \beta_s|$. Whenever some $\beta_s$ is close to 1, $\mathbf{H}_s$ is nearly singular (and exactly singular if $\beta_s = 1$), driving $\sigma_{\min}(\mathbf{P}_{j \to i})$ toward zero. Volume contraction is quantified by

$$\det(\mathbf{P}_{j \to i}) = \prod_{s=j+1}^{i} (1 - \beta_s),$$

which typically decays exponentially in $i - j$ unless $\beta_s$ concentrates at the orthogonal endpoints $\{0, 2\}$.

**Aligned-plane special case.** If the directions are time-invariant, $\mathbf{w}_s \equiv \mathbf{w}$, then $\mathbf{P}_t = \mathbf{w}\mathbf{w}^\top$ is an idempotent projector and the factors commute:

$$\prod_{s=j+1}^{i} \mathbf{H}_s = \prod_{s=j+1}^{i} (\mathbf{I} - \beta_s \mathbf{P}) = \mathbf{I} - \Big(1 - \prod_{s=j+1}^{i} (1 - \beta_s)\Big)\mathbf{P},$$

so the eigenvalue along $w$ is exactly $\prod_{s=j+1}^{i}(1 - \beta_s)$, making the contraction along $w$ explicit and exponential in path length unless $\beta_s \in \{0, 2\}$.

**Implications for long-context modeling.** Because the PaTH transport multiplies the Q/K bilinear by $\mathbf{P}_{j \to i}$, any persistent deviation of $\beta_t$ from $\{0, 2\}$ yields cumulative energy loss along a moving one-dimensional subspace. This concentrates mass in progressively fewer directions and can flatten or attenuate long-range logits $\mathbf{q}_i^\top \mathbf{P}_{j \to i} \mathbf{k}_j$ as $i - j$ grows, unless additional renormalizations or forget-gates are introduced. In contrast, **GRAPE-M** maps lie in $SO(d)$, so for both non-contextual and contextual types, all singular values are 1; volumes and norms are preserved, and Lyapunov exponents are 0, avoiding contraction-induced degradation of long-range interactions.

**Lemma J.7** (Orthogonality condition for PaTH factors). For $\mathbf{H}_t = \mathbf{I} - \beta_t \mathbf{w}_t \mathbf{w}_t^\top$ with $\|\mathbf{w}_t\| = 1$, $\mathbf{H}_t$ is orthogonal iff $\beta_t \in \{0, 2\}$. For $\beta_t \in (0, 2) \setminus \{0, 2\}$, $\mathbf{H}_t$ is symmetric, diagonalizable with eigenvalues in $(-1, 1] \cup \{1\}$, and strictly contractive on $\mathrm{span}\{\mathbf{w}_t\}$.

