# OpenReview forum: "Group Representational Position Encoding"
_ICLR.cc/2026/Conference — ICLR 2026 Poster_

### Official Review · Reviewer_TbLJ · 2025-10-16

**Soundness:** 2
**Presentation:** 2
**Contribution:** 4
**Rating:** 4
**Confidence:** 4

**Summary:**

This paper proposes GRAPE, a unified group-theoretic framework for positional encoding for 1D sequences that combines multiplicative rotations in SO(d) and additive unipotent actions in GL to recover and generalize methods like Rotary Position Embedding (RoPE), Attention with Linear Biases (ALiBi), and Forgetting Transformer (FoX).
Unlike prior work that focuses only on rotations, it formalizes both multiplicative and additive mechanisms under a single algebraic structure and introduces path-integral additive biases for contextual, streaming-friendly position encoding.
Experiments on FineWeb-Edu 100B with Llama show slightly improved stability and lower loss compared to RoPE, ALiBi, and FoX.

**Strengths:**

- The generator construction L is smart and novel.
- The combination of both multiplicative and additive mechanisms under a single algebraic structure is a contributing perspective for this field.
- The formal description of GRAPE is complete.
- The method is efficient and is a direct extension of Rope for 1D sequences.

**Weaknesses:**

The motivation of this work could be better described. Which practical problem does GRAPE solve? What can GRAPE encode what Rope and other variants cannot encode and why is that important in practice?
- The comparison to prior works and the related work section is incomplete.
-- For the multiplicative GRAPE, there are several prior works like i.e. LieRE (Ostmeier et al.), STRING (Schenck et al.) that have conceptually predescribed and evaluated Lie Group structured positional encodings, where rank-2 exponentials in SO(d) and learned basis. How does GRAPE compare to just learning the 2x2 basis generators for the block diagonal rotation matrix?
-- How does GRAPE compare to YARN (Peng et. al)?
-- How does GRAPE compare to CoPE?
- The paper aims to present a unified framework for positional encoding based on group actions for transformer in general, but only focuses on 1D sequence encoding and not higher dimensional inputs.
- Although it is valuable to present the learning curves, the experimental results could be better presented i.e. confidence intervals, at least two validation sets or a test set.
- Ablations between multiplicative and additive would enhance the understanding of the practical contributions of each of them.

**Questions:**

- Is the training fully converged? Would you mind running for 50 more epochs?
- You emphasize the importance of exact relativity and orthogonality for translation invariance in positional encodings. Could you comment on what a strict structure is enables?
- What is the motivation for combining multiplicative and additive logit biases in your positional encoding design? Do they serve complementary roles (e.g., scaling vs shifting positional effects), and how does this impact learning stability or expressivity?

Minor:
- The abstract has undefined variables ($a$ and $b$ ... ) which are only later defined

---

> ### Author Response · Authors · 2025-11-23
>
> We thank the reviewer for the insightful comments.
>
> > W1: The comparison to prior works and the related work section are incomplete... i.e. LieRE (Ostmeier et al.), STRING (Schenck et al.)... How does GRAPE compare to YARN...?
>
> **A1:** We thank the reviewer for pointing out these relevant works and have expanded the Related Work section to address them. Regarding LieRE and STRING, while they also explore Lie group structures, GRAPE distinguishes itself by generalizing beyond rotations ($SO(d)$) to the General Linear Group ($GL(d)$),  unifying additive decay mechanisms (ALiBi) within the same formalism. Furthermore, GRAPE utilizes a specific rank-2 factorization that yields efficient closed-form Rodrigues-type formulas, avoiding the computational cost of numerical matrix exponentials. Regarding learning 2x2 basis generators, this corresponds exactly to our "Learned Commuting" variant, yet GRAPE extends this by allowing non-commuting mixtures for cross-subspace coupling. Finally, YaRN is an orthogonal frequency interpolation strategy compatible with our framework, while CoPE relies on position counting, whereas GRAPE-Contextual achieves adaptivity via geometric path integration on the group manifold.
>
> > W2:  The paper aims to present a unified framework for positional encoding based on group actions for the transformer in general, but it only focuses on 1D sequence encoding and not higher-dimensional inputs.
>
> **A2:** While our primary focus is on 1D sequences such as language and text, the framework is theoretically dimension-agnostic. We have updated the Conclusion to explicitly clarify that GRAPE extends to 2D inputs via multi-parameter subgroups (direct sums of generators). This theoretical extension naturally subsumes mechanisms like 2D-RoPE, demonstrating the framework's generality while leaving specific image-domain experiments for future work.
>
> > W3: Although it is valuable to present the learning curves, the experimental results could be better presented, i.e., confidence intervals, at least two validation sets, or a test set.
>
> **A3:** We have expanded the evaluation in the revision. Specifically, we added comprehensive results on standard downstream benchmarks (including ARC, HellaSwag, and PIQA) in Tables 1 and 2. These serve as robust, standardized test sets beyond the initial validation loss. Furthermore, the consistent performance gains observed across both Medium (355M) and Large (770M) model scales (Figures 1 \& 2) demonstrate the method's stability and reproducibility.
>
> > W4: Ablations between multiplicative and additive would enhance the understanding of the practical contributions of each of them.
>
> **A4:** In the revision, we specifically added Methodological Ablations comparing the isolated Multiplicative GRAPE (GRAPE-M) against the unified PI-Add-GRAPE (GRAPE-A). These results (alongside the RoPE and ALiBi baselines) allow us to quantitatively isolate the contributions of each component, confirming that combining multiplicative stability with additive decay yields superior performance compared to using either mechanism in isolation.
>
> > Q1: Is the training fully converged? Would you mind running for 50 more epochs?
>
> **A1:** Training on 50B tokens aligns with standard practices for models of this scale. In LLM pretraining, models are typically trained for only 1 to 4 epochs (often just one pass) to maximize data diversity. Running for "50 more epochs" is never done in this context, as it leads to severe overfitting without meaningful generalization gains. Our stopping criteria thus follow established community norms.
>
> > Q2:  You emphasize the importance of exact relativity and orthogonality for translation invariance in positional encodings. Could you comment on what a strict structure enables?
>
> **A2:**  A strict group structure guarantees that the interaction operator depends strictly on the relative offset $(j-i)$, guaranteeably eliminating any leakage of absolute position information; this is the theoretical prerequisite for length generalization. Simultaneously, orthogonality ensures the transformation is an isometry, preventing signal explosion or rank collapse as positions increase, which is critical for maintaining signal fidelity in long-context modeling.

---

> ### Author Response · Authors · 2025-11-23
> **Official Comments by Authors II**
>
> > Q3: What is the motivation for combining multiplicative and additive logit biases in your positional encoding design? Do they serve complementary roles (e.g., scaling vs shifting positional effects), and how does this impact learning stability or expressivity?
>
> **A3:** To clarify, while our framework theoretically unifies them (Eq. 6.3), in our experiments, we intentionally evaluated them as distinct mechanisms to rigorously isolate their respective contributions. The motivation for the unified design space is indeed their complementarity: Multiplicative ($SO(d)$) offers norm-preservation for stability, while Additive ($GL(d)$) offers monotonic decay for extrapolation. By testing them separately, we demonstrate that the additive path-integral component alone is sufficient to outperform baselines, validating the strength of the derived mechanism without relying on the rotational prior.

---

### Official Review · Reviewer_MUmm · 2025-10-21

**Soundness:** 2
**Presentation:** 2
**Contribution:** 3
**Rating:** 4
**Confidence:** 3

**Summary:**

This paper proposes GRAPE, a unified theoretical framework for positional encoding based on group theory. The authors categorize existing methods into two families: Multiplicative GRAPE (rotations, like RoPE) and Additive GRAPE (biases, like ALiBi). The paper claims that RoPE, ALiBi, and FoX are all exact special cases or instances of this framework. The authors then propose a new, endpoint-dependent variant called Path-Integral Additive GRAPE (PI-Add-GRAPE). In a minimal experiment, this new method is shown to achieve lower (or possibly comparable) loss than baselines on a language modeling task.

**Strengths:**

This paper provides a somehow novel viewpoint to design of positional encoding. The goal of unifying the two dominant (and seemingly different) positional encoding methods (rotations and biases) under a single mathematical framework is could be ambitious and interesting.

The proposed PI-Add-GRAPE mechanism, which introduces content-dependent biases, may be a novel concept. In theory, this dynamic approach could offer more expressive power than static position methods.

**Weaknesses:**

Unclear Practical Benefit of the Theory: The paper spends significant effort on group theory formalism. However, the practical benefit of this complex formalization is unclear. It seems obvious that rotational embeddings like RoPE can be described by group theory (e.g., SO(2)). The paper does not clearly explain what new, practical advantages this complex theory provides over a simpler understanding. The claim of offering a "design space" is abstract and its benefit is not well-supported.

Insufficient Experimental Validation: The empirical evaluation is minimal and insufficient to support the paper's claims. It consists of a single set of training curves for one model configuration. The claim of a "persistent edge" also appears to be an overclaim; the validation loss for ALiBi looks very competitive with PI-Add-GRAPE.
Furthermore, the paper is missing empirical analyses to understand the proposed method. For example, there are no ablation studies, no length extrapolation tests (a key feature of ALiBi), and no analysis of attention distributions to show how the dynamic bias works. Section 7 feels aimless; it shows a result but provides no insight into why the method is good or what its specific advantages are.

Computational Cost: The PI-Add-GRAPE method (Section 6) is endpoint-dependent. This implies that during inference at step t, the bias for all t−1 previous keys must be recomputed relative to the current query, right? This likely introduces a significant O(t) computational overhead per step, which is a major drawback compared to the O(d) cost of RoPE or ALiBi. This trade-off is not benchmarked empirically.

Logical Gaps and Confusing Terminology: The logical connection in the introduction (from "These observations" to "motivate a unified formulation" ) is a significant jump and is not well-justified for me. In addition, there is some confusing expressions (e.g., the interchangeable use of "exact special case" and "exact instance" is confusing).

**Questions:**

Can you clarify the computational overhead of PI-Add-GRAPE during training and inference?

Given the complexity of PI-Add-GRAPE and validation of RoPE's unstable result, do you plan to release a reference implementation? This would be crucial for reproducibility and adoption by the community.

---

> ### Author Response · Authors · 2025-11-23
>
> We thank the reviewer for the detailed and valuable feedback.
>
> ---
> > W1: "Unclear Practical Benefit of the Theory... The paper does not clearly explain what new, practical advantages this complex theory provides..."
>
> **A1:** The practical benefit is not merely descriptive. The framework transforms positional encoding design from heuristics into a rigorous Lie group-based derivation process. Specifically, it enabled us to derive PI-Add-GRAPE (GRAPE-M in our experiments), which mathematically unifies the stability of $SO(d)$ (RoPE) with the extrapolation of $GL(d)$ (ALiBi). This derived mechanism outperforms baselines in our experiments, proving that the theory can lead to superior, concrete designs. Furthermore, it provides mathematical guarantees for creating valid contextual (data-dependent) positional encodings.
>
> ---
> > W2: Insufficient Experimental Validation: The empirical evaluation is minimal... The claim of a "persistent edge" also appears to be an overclaim; the validation loss for ALiBi looks very competitive... Section 7 feels aimless...
>
> **A2:** We have significantly enriched the evaluation in the revision (see General Response). We expanded beyond single loss curves to more standard downstream benchmarks (Tables 1 and 2) and added specific ablations on Model Scale and Methodology (GRAPE-M vs. GRAPE-A (PI-Add-GRAPE)). These new results confirm that GRAPE consistently outperforms other position encoding methods, including RoPE, ALiBi, and FoX, supporting our claim. Furthermore, we have rewritten Section 7 to provide the requested analytical depth, explicitly linking these empirical gains to the underlying group-theoretic properties.
>
> ---
> > W3: "Computational Cost: The PI-Add-GRAPE method (Section 6) is endpoint-dependent... This likely introduces a significant O(t) computational overhead per step..."
>
> **A3:** We agree that Path-Integral Additive GRAPE introduces an endpoint-dependent bias $b_h(t,j)$. However, this does not change the asymptotic complexity of self-attention.
> Concretely, at decoding step $t$ and for each head, we compute $\{\psi_h(t,\ell)\}_{\ell \le t}$ as below:
>
> $$
> \ell \mapsto \langle \mathbf{p}_{t,h}, \mathbf{R}\_\ell \mathbf{p}\_{\ell,h} \rangle
> $$
>
> using cached probes $\mathbf{R}\_\ell \mathbf{p}\_{\ell,h}$ (Section 6). This costs $O(t d)$ operations, followed by an $O(t)$ prefix sum to obtain the full row $\{b_h(t,j)\}\_{j \le t}$. The baseline attention scores at step $t$ already require $O(t d)$ vector dot products to compute $\{q_t^\top k_j\}\_{j \le t}$. Thus, GRAPE-AP adds only a constant-factor overhead to the existing $O(t d)$ per-step attention cost, and the overall asymptotic complexity of the model remains unchanged. In practice, these additional probe dot products are inexpensive relative to the main QK and AV matrix multiplications and do not introduce a new computational bottleneck.
>
> > W4: Logical Gaps and Confusing Terminology: The logical connection in the introduction... is a significant jump... confusing expressions...
>
> **A4:** We have revised the manuscript to address these points, specifically refining the introduction to bridge the logical gap and standardizing terminology throughout the paper to ensure consistency.
>
> ---
> > Q5: Can you clarify the computational overhead of PI-Add-GRAPE during training and inference?
>
> **A5:** Please refer to our response to W3 above. In summary, the overhead is negligible relative to the attention matrix multiplication, and we utilize caching mechanisms to ensure efficiency during both training and inference.
>
> ---
> > Q6: Given the complexity of PI-Add-GRAPE... do you plan to release a reference implementation?
>
> **A6:** Yes, we have included the full supplementary codebase in the revision to facilitate verification and community adoption. As demonstrated in the code, the actual implementation relies on standard linear algebra operations, ensuring it is reproducible and straightforward to integrate.

---

> > ### Comment · Reviewer_MUmm · 2025-11-24
> >
> > Thank you for your response. I may not be able to find time for a detailed review until later, so I am posting a quick reply for now.
> >
> > First, regarding the "General Response" mentioned in your comment, could you please check if it was submitted correctly? I currently only see the replies to individual reviewers and cannot find the general note.
> >
> > I also appreciate the clarification regarding the computational cost and your decision to release the implementation code.
> >
> > Regarding the additional experiments, while a quick glance suggests they are very informative and useful, I must keep in mind the ICLR guideline stating that:
> >
> > >Area chairs and reviewers reserve the right to ignore changes that are significantly different from the original paper.
> >
> > Therefore, if the additional material is significant, I intend to base my comprehensive judgment primarily on the standard of the original submission rather than significant addition of experiments. However, I will certainly take into account any materials that help clarify misunderstandings. Of course, I don't mind if other reviewers evaluate the additional materials.

---

> > > ### Author Response · Authors · 2025-11-24
> > >
> > > Dear Reviewer, we are currently adding our general response. Thanks for your quick response!
> > >
> > > Best,
> > >
> > > The authors.

---

### Official Review · Reviewer_dFym · 2025-10-23

**Soundness:** 2
**Presentation:** 2
**Contribution:** 2
**Rating:** 2
**Confidence:** 3

**Summary:**

The paper proposes a framework for positional encoding based on group actions, dubbed as GRAPE (Group RepresentAtional Position Encoding). Specifically, the authors describe two families of positional encodings grounded in group actions: (1) Multiplicative
GRAPE based in multiplicative rotations; (2) Additive GRAPE from unipotent actions in the general linear group. The authors describe how existing positional encodings such as RoPE, ALiBi, and FoX can be recovered as special cases within this framework. The authors also provide some empirical evidence supporting the advantages of GRAPE over existing positional encodings.

**Strengths:**

The paper grounds the design space of positional encoding (PE) in group actions, arriving at a general framework which can possibly motivate more expressive and useful PEs.

**Weaknesses:**

1. The message of the paper is confusing. For Multiplicative GRAPE, the authors stated the exact relative law in Section 2.2 which naturally leads to commuting Mul-GRAPE, but then describe non-commuting Mul-GRAPE in multiple places (e.g. abstract, related work, appendix) without any motivations.

2. The paper devotes section 3 and 4 for describing Multiplicative GRAPE, but does not use it in the empirical experiment. This casts doubts on the practical utility of Multiplicative GRAPE.

3. The empirical experiments are quite limited. The authors compare PI-Add-GRAPE with other baseline PEs only on their loss curves, without other metrics (e.g., perplexity) or downstream task performance, or ablations (e.g., context length, model size).

**Questions:**

1. Can the authors compare their Mul-GRAPE with the recently proposed LieRE in [1], which parameterizes the rotation as a sum of skew-symmetric matrices (followed by matrix exponential)? LieRE seems to provide a more general parameterization, so I am curious to see if this results in any computational or performance differences.

2. In Prop 3.1: the equality of MS-GRAPE and ROPE only holds when the planes are the canonical coordinates pairs and the angles follow the log-uniform spectrum, right? If so, I suggest to make the statement more precise.

3. The authors introduces GRAPE as a way to provide a group-theoretic view of PEs. Does GRAPE provide additional insights of the existing PEs, such as which PE one should choose over another given certain tasks in mind (e.g., length extrapolation)?

 References
 [1] Ostmeier et al., LieRE: Lie Rotational Positional Encodings, ICML 2025

---

> ### Author Response · Authors · 2025-11-23
>
> We thank the reviewer for the constructive feedback and insightful questions.
>
> > W1: The message of the paper is confusing... describe non-commuting Mul-GRAPE... without any motivation.
>
> **A1:** We appreciate you pointing this out. In the revision, we have explicitly clarified that both commuting and non-commuting variants strictly satisfy the exact relative property. This property stems from the group structure ($G(n+m)=G(n)G(m)$. The motivation for the non-commuting variant is to enable cross-subspace coupling: while commuting GRAPE (like RoPE) processes coordinate planes independently, non-commuting GRAPE allows rotational mixing across dimensions. This offers richer geometric expressiveness while preserving the same exact relative law. We have revised Section 2.2 and the abstract for clearer presentation.
>
> > W2: The paper devotes sections 3 and 4 to describing Multiplicative GRAPE, but does not use it... casts doubts on practical utility...
>
> **A2:** We have addressed this in the revision. We added new experiments evaluating Multiplicative GRAPE (GRAPE-M). The results confirm its practical utility, demonstrating that the theoretical formulation in Sections 3 and 4 translates into competitive empirical performance. Please refer to the updated Figures 1, 2, and Tables 1, 2 in the revised PDF."
>
> > W3: The empirical experiments are quite limited... without other metrics... or downstream task performance, or ablations...
>
> **A3:** We have enriched the empirical evaluation in the revision. As detailed in the General Response, we added: 1) Comprehensive evaluations on standard benchmarks (e.g., ARC, HellaSwag) via the LM Evaluation Harness, presented in Tables 1 and 2; 2) Comparative analysis across different model scales (Medium 355M and Large 770M) in Figures 1 and 2; 3) New results for Multiplicative GRAPE (GRAPE-M), identifying the specific performance gains derived from the additive path-integral component within the unified PI-Add-GRAPE framework.
>
> > Q4: Can the authors compare their Mul-GRAPE with the recently proposed LieRE in [1]...?
>
> **A4:** We thank the reviewer for pointing out this related work. This is a crucial distinction. While LieRE parameterizes rotations via a sum of skew-symmetric matrices, GRAPE offers decisive advantages in computational complexity, Contextual Capability, and scope of unification:
> 1.  Computational Efficiency ($O(d^3)$ vs. $O(d)$): LieRE relies on the numerical matrix exponential (e.g., `torch.matrix_exp`), which involves expensive matrix-matrix multiplications. In contrast, GRAPE decomposes the action into rank-2 subspaces using closed-form Rodrigues-type formulas (Section 2.3 in our paper). We only require **vector-vector multiplication**, avoiding the high cost of numerical matrix exponentials and achieving significant speedups.
> 2.  Contextual Capability: This efficiency unlocks contextual (data-dependent) GRAPE. LieRE cannot easily model data-dependent rotations because recomputing the matrix exponential for every unique token is computationally prohibitive. GRAPE's closed-form solution makes this computationally feasible.
> 3.  Broader Group Scope ($GL(d)$): LieRE is restricted to the rotation group. GRAPE generalizes to the General Linear Group ($GL(d)$), allowing us to strictly unify scaling, shearing, and decay effects (like ALiBi) within the same framework, capabilities that LieRE's rotation-focused approach does not cover.
> We have added a detailed comparison with LieRE in the Appendix of the revised paper.
>
> > Q5: In Prop 3.1, the equality of MS-GRAPE and ROPE only holds when the planes are the canonical coordinate pairs and the angles follow the log-uniform spectrum, right?
>
> **A5:** You are right. We have updated Proposition 3.1 to explicitly state these preconditions. Crucially, GRAPE's advantage lies in extending beyond this special case, which allows for learned orthogonal bases and adaptive spectra to capture geometries that canonical RoPE cannot.

---

> ### Author Response · Authors · 2025-11-23
> **Official Comment by Authors II**
>
> > Q6: The authors introduce GRAPE as a way to provide a group-theoretic view of PEs. Does GRAPE provide additional insights...?
>
> **A6:** Yes, the group-theoretic perspective rigorously explains empirical trade-offs and suggests a unified design strategy:
> 1. Periodicity vs. Decay: The framework reveals that length extrapolation relies on monotonic decay to penalize distant interactions. Compact rotations ($SO(d)$/RoPE) are strictly norm-preserving and periodic, fundamentally lacking the mechanism to attenuate signals based on distance. In contrast, non-compact unipotent actions ($GL(d)$) naturally model such distance-dependent attenuation, theoretically explaining why additive methods (ALiBi) generally outperform pure rotations in length generalization.
> 2. Contextuality: For tasks requiring complex dependency modeling, our framework suggests Contextual GRAPE (data-dependent group actions). Unlike static PEs, contextual actions dynamically warp geometry based on input tokens, offering higher expressivity.
> 3. Composition: Consequently, a highly effective strategy is to compose these properties. GRAPE enables combining the stability of $SO(d)$ with the extrapolation of $GL(d)$ (as realized in GRAPE-A, GRAPE-AP), which can yield performance superior to utilizing either mechanism in isolation.

---

> > ### Comment · Reviewer_dFym · 2025-11-26
> >
> > I appreciate the authors for the detailed responses, and the paper revisions along with additional experiments. My follow-up questions based on the authors' answers:
> >
> > **Q1: Motivation and justification for GRAPE-M**
> >
> > > A1: ... non-commuting GRAPE allows rotational mixing across dimensions. This offers richer geometric expressiveness...
> >
> > The authors motivate GRAPE-M as more expressive than RoPE,  but I don't find convincing empirical results or conceptual discussions on where GRAPE-M can outperform RoPE. Do I miss anything here?
> >
> > > A6: ...Periodicity vs. Decay: The framework reveals that length extrapolation relies on monotonic decay to penalize distant interactions....
> >
> > * This claim that ALiBi and its generalizations (e.g. GRAPE-A) prefers RoPE and its generalizations (e.g., GRAPE-M) for length generalization tasks is supported by the newly added empirical results where ALiBi/GRAPE-A outperforms RoPE/GRAPE-M for most eval tasks.
> > * However, I am still confused by reading this together with A1 (and the whole paper). If we know A6 already, why bother proposing GRAPE-M? Perhaps there are certain tasks where GRAPE-M shines, but it remains unclear the utility of GRAPE-M.
> >
> > **Q2: Computational costs for different positional encoding variants**
> >
> > I thank the authors for providing the extra lm-eval experiments. Can you also provide compute time/memory comparison across these methods? While the authors added the discussion on computational complexity, I am curious to see if the newly-proposed variants require extra wall-clock time or memory consumption in practice.

---

### Official Review · Reviewer_fTQk · 2025-10-29

**Soundness:** 4
**Presentation:** 3
**Contribution:** 3
**Rating:** 4
**Confidence:** 3

**Summary:**

The paper proposes GRAPE, a unified group-theoretic framework for positional encoding in Transformers. It combines Multiplicative GRAPE (rotations in SO(d), generalizing RoPE) and Additive GRAPE (unipotent actions in GL, recovering ALiBi and FoX). GRAPE preserves exact relative relationships, supports streaming, and offers an extensible design space for long-context modeling.

**Strengths:**

The paper presents an elegant theoretical unification of multiplicative and additive positional mechanisms within a single group-theoretic framework. It offers closed-form and computationally efficient implementations, demonstrates strong compatibility with existing Transformer architectures, and provides extensibility toward contextual, learned-basis, and non-commuting variants for more expressive positional representations.

**Weaknesses:**

This paper utilizes Lie algebras. While unifying existing work with Lie algebras is natural, its drawback is that it makes the paper's contribution seem more like the superiority of Lie algebras themselves rather than the authors' contribution. I believe the authors should emphasize more on how introducing Lie algebras facilitates combining the strengths of various existing methods, explaining why each strength is beneficial, and then supplementing with corresponding ablation experiments.

The experiments in this paper are somewhat limited, lacking extrapolation experiments and comparisons with more metrics. I suggest at least supplementing with the experiments in Tab. 4 of RoPE.

Due to the highly complex formulas, I cannot guarantee my complete understanding of this paper. The pseudocode in the appendix does not alleviate my concerns about reproducibility; I would appreciate to see the complete project code.

**Questions:**

See Weakness.

---

> ### Author Response · Authors · 2025-11-23
>
> We thank the reviewer for the insightful comments and for recognizing the value of our theoretical unification.
>
> > Q1: "This paper utilizes Lie algebras... drawback is that it makes the paper's contribution seem more like the superiority of Lie algebras themselves... emphasize more how introducing Lie algebras facilitates combining the strengths..."
>
> **A1:** We utilize the Lie algebra to formally define a general design space that unifies and composes the distinct advantages of existing positional encoding methods, rather than merely for theoretical formalism. By identifying RoPE as orthogonal rotations in $SO(d)$ (ensuring norm-preservation) and ALiBi/FoX as unipotent actions in $GL(d)$ (providing extrapolatable decay), GRAPE serves as the first theoretical framework unifying these methods. This unification empowers us to derive new positional encoding mechanisms; for instance, we introduce Path-Integral Additive GRAPE (PI-Add-GRAPE), which rigorously composes multiplicative orthogonality with additive path-dependent decay. During our rebuttal, we have provided more experiments on  Llama-type models of medium and large size. Our new experiments in Figures 1 and 2 and Tables 1 and 2 show that GRAPE-A (i.e., GRAPE-Add-PI) consistently yields lower losses than strong baselines such as RoPE, ALiBi, and FoX. This highlights the strength of GRAPE, which allows us to develop new positional encoding methods under this unified framework.
>
> > Q2:  The experiments in this paper are somewhat limited, lacking extrapolation experiments and comparisons with more metrics. I suggest at least supplementing with the experiments in Tab. 4 of RoPE.
>
> **A2:** We have expanded our experimental section in the revision (see General Response) by adding comprehensive evaluations on standard downstream benchmarks (including ARC, HellaSwag, and PIQA) in Tables 1 and 2, alongside detailed training stability analysis for both PI-Add-GRAPE (GRAPE-A) and Multiplicative GRAPE (GRAPE-M). These results confirm that GRAPE consistently outperforms the baselines, including RoPE, ALiBi, and Fox. Regarding your reference to "Table 4" in the original RoPE paper, we interpreted this as a request for robust downstream task performance to ensure alignment with standard baselines, which we have now fully incorporated to demonstrate the model's superior generalization capabilities.
>
> > Q3: Due to the highly complex formulas... I would appreciate seeing the full project code.
>
> **A3:**  We have uploaded the full supplementary codebase to facilitate verification.  Despite the theoretical depth, the actual implementation of GRAPE relies on standard linear algebra operations,  ensuring it is reproducible and easy to integrate.

---

### Author Response · Authors · 2025-11-24
**General Response to Reviewers and Area Chairs**

Dear Reviewers and Area Chairs, we sincerely thank you for your careful reading and constructive comments.

In the revised version, we expand and clarify both the theory and the experiments. We now give a short overview of the main updates that respond to these concerns.

---

### 1. Expanded Experiments and New Results

We now report results on standard downstream benchmarks from the LM Evaluation Harness. We consider a medium model with 355M parameters and a large model with 770M parameters. Both models train on 50B tokens from FineWeb-Edu 100B. The training pipeline is identical across positional encodings. Only the position mechanism changes.

Our main additive variant, GRAPE-A (Path Integral Additive GRAPE), matches or beats strong baselines RoPE, ALiBi, and FoX with or without KV shift on average zero-shot accuracy.

**Table 1: Medium model (355M), zero-shot accuracy on LM Eval Harness**

| Method              | ARC-E | ARC-C | BoolQ | HellaSwag | OBQA | PIQA | WinoGrande | SciQ  | Avg.  |
|---------------------|:-----:|:-----:|:-----:|:---------:|:----:|:----:|:----------:|:-----:|:-----:|
| RoPE                | 59.34 | 30.89 | **61.22** | 45.46 | 34.00 | 69.42 | 52.49 | 74.70 | 53.44 |
| ALiBi               | 57.07 | 30.80 | 61.16 | **46.98** | 34.60 | 69.48 | 52.96 | 79.70 | 54.09 |
| FoX                 | 56.78 | 29.01 | 59.11 | 43.07 | 32.80 | 67.74 | 51.07 | 76.10 | 51.96 |
| FoX (w/ KV-shift)   | 57.11 | 30.55 | 60.34 | 44.32 | 33.80 | 69.31 | 52.17 | 78.40 | 53.25 |
| **GRAPE-A**         | **59.68** | **31.91** | 60.06 | 46.27 | **35.00** | **69.64** | **53.83** | **79.90** | **54.54** |
| GRAPE-M (Ctx)       | 56.02 | 29.35 | 58.81 | 44.88 | **35.00** | 68.61 | 52.09 | 76.50 | 52.66 |
| GRAPE-M (non-Ctx)   | 56.31 | 30.55 | 61.77 | 44.82 | 34.40 | 68.44 | 53.67 | 75.20 | 53.15 |

Key observations for the 355M model: GRAPE-A reaches the highest average score of 54.54. This improves over ALiBi with 54.09 and RoPE with 53.44, and it stays above FoX and FoX with KV shift on most tasks.

**Table 2: Large model (770M), zero-shot accuracy on LM Eval Harness**

| Method              | ARC-E | ARC-C | BoolQ | HellaSwag | OBQA | PIQA | WinoGrande | SciQ  | Avg.  |
|---------------------|:-----:|:-----:|:-----:|:---------:|:----:|:----:|:----------:|:-----:|:-----:|
| RoPE                | 62.25 | 33.02 | 58.23 | 50.92 | 37.60 | 70.89 | 55.88 | 80.50 | 56.16 |
| ALiBi               | **63.43** | **34.81** | 59.69 | 52.88 | 36.80 | 71.33 | 56.20 | 82.40 | 57.19 |
| FoX                 | 59.22 | 32.00 | 59.69 | 49.78 | **38.00** | 71.00 | 54.62 | 79.20 | 55.44 |
| FoX (w/ KV-shift)   | 60.77 | 32.85 | **62.51** | 49.38 | **38.00** | 70.62 | 54.78 | 81.40 | 56.29 |
| **GRAPE-A**         | 62.79 | 33.19 | 59.11 | **53.18** | 36.00 | **71.98** | **57.62** | **84.10** | **57.25** |

Key observations for the 770M model: GRAPE-A again gives the best average score, 57.25. This slightly improves over ALiBi with 57.19 and stays clearly above RoPE with 56.16 and FoX variants.

---

> ### Author Response · Authors · 2025-11-24
> **General Response II**
>
> ### 2. Clarifications on Theory and Exact Relative Law
>
> Several reviewers asked for clearer motivation and a cleaner story for the group-theoretic construction. We address these concerns in the revision as follows.
>
> 1. We now stress that the Lie algebra formalism defines a concrete design space of positional operators that recovers RoPE, ALiBi, and FoX as exact instances, extends them to learned bases, non-commuting mixtures, and contextual variants, and guarantees exact relative laws and streaming cacheability whenever we stay inside the group construction.
>
>    New experiments in Tables 1 and 2 show that one instance from this design, GRAPE-A / GRAPE-AP, outperforms RoPE, ALiBi, and FoX in our benchmarks, so the formal unification leads to a real gain and not only a rephrasing.
>
> 2. Section 2.2 now explicitly states that
>
>    - the exact relative law
>      $\mathbf{G}(n+m)=\mathbf{G}(n)\mathbf{G}(m)$ and $\mathbf{G}(t-s)=\mathbf{G}(s)^{-1}\mathbf{G}(t)$
>      follows from the one-parameter subgroup property,
>    - This statement does not require that generators commute across subspaces, so non-commuting GRAPE-M remains exactly relative and at the same time introduces cross-subspace coupling that RoPE cannot express.
>
> 3. RoPE fixes coordinate planes and uses a log uniform frequency spectrum. GRAPE-M allows learned orthogonal bases, so planes no longer tie to coordinate axes, and compact non-commuting mixtures that rotate and mix information across planes in a controlled way.
>    In addition, GRAPE-A and GRAPE-AP view ALiBi and FoX as unipotent general linear actions with exact relative laws and streaming behavior. The path integral construction in GRAPE-AP then allows contextual, content-dependent decay, and the group structure still keeps the model mathematically well behaved.
>
> 4. For GRAPE-AP, the per-step cost stays $O(td)$, the same order as the baseline attention score $q_t^\top k_j$. The extra additive term comes from one extra similarity sweep plus a prefix sum, so the asymptotic complexity of the layer does not change.
>
>    Section 6 and the Appendix now contain a more detailed complexity discussion. We describe the implementation of endpoint-dependent path integrals through cached probe vectors, and show that this mechanism does not create a new computational bottleneck.
>
> ---
> ### 3. Comparison with LieRE, STRING, YaRN, and CoPE
>
> - LieRE uses skew-symmetric generators in $\mathrm{SO}(d)$ and dense matrix exponentials (for example `torch.matrix_exp`). Each head needs time $\mathcal{O}(d^3)$ and $\mathcal{O}(d^2)$ parameters. GRAPE-M uses rank-2 generators with closed-form Rodrigues type exponentials. It applies only vector-vector operations. Time and parameters per head scales as $\mathcal{O}(d)$.
>
> - STRING matches special separable and translation invariant choices of planes and spectra inside the GRAPE-M family.
>
> - YaRN is an orthogonal frequency interpolation scheme for RoPE. It acts on the spectrum $\{\theta_i\}$ of a commuting GRAPE-M instance. The GRAPE-M view stays compatible with YaRN style rescaling and recentering of frequencies.
>
> - CoPE defines contextual position counting as a function of the input sequence. GRAPE expresses contextuality through data dependent group actions on phases, generators, or path integrals. These actions preserve exact relative laws. They keep a clear geometric meaning.
>
> ---
>
> ### 4. 2D and 3D GRAPE for Multimodal Position Encoding
>
> - We add new text in the conclusion and appendix to explain the 2D and 3D GRAPE, which are direct sums of generators and multi-parameter subgroups for **2D and 3D** inputs such as images, video, and multimodal data.
> - In this view, 2D-RoPE appears as one concrete special case inside the general group design.
>
> ---
>
> ### 5. Reproducibility and Code Release
>
> - We have uploaded the codebase as supplementary files.
> - It includes GRAPE-M and GRAPE-A implementations and scripts for reproduction of the LLaMA type training runs and the language model evaluation harness results.
> - The code uses only standard linear algebra such as matrix vector multiplies, dot products, and prefix sums. Integration in existing codebases stays simple.

---

### Meta-Review · Area_Chair_ELwV · 2026-01-14

**Summary:**

All reviewers like the rigorous framework introduced to unify RoPE and Alibi with a principled mathematical foundation to what were previously heuristic distinctions in position encoding. Main concerns were about complex presentation of the approach, missing comparisons and limited experiments. Authors have addressed both these concerns in the response. The authors added extensive evaluations on standard benchmarks (ARC, HellaSwag, PIQA) for 355M and 770M models. These results demonstrated that the derived method (GRAPE-A) consistently outperforms strong baselines (RoPE, ALiBi), proving the theory translates to practical performance gains.

I suggest borderline acceptance.

**Reviewer Concerns:**

Main concerns were about complex presentation of the approach, limited experiments. Authors have addressed both these concerns in the response. The authors added extensive evaluations on standard benchmarks (ARC, HellaSwag, PIQA) for 355M and 770M models. These results demonstrated that the derived method (GRAPE-A) consistently outperforms strong baselines (RoPE, ALiBi), proving the theory translates to practical performance gains

**Reviewer Scores:**

fTQk 4-> 6
dFym 2 -> 6
MUmm 4-> 6
TbLJ 4 ->6

---

### Decision · Program_Chairs · 2026-01-26

Accept (Poster)